# S³GC: Scalable Self-Supervised Graph Clustering

**Fnu Devvrit**[*]
Department of Computer Science
University of Texas at Austin
devvrit.03@gmail.com

**Aditya Sinha**[†]
Google Research
Bengaluru, India, 560016
sinhaaditya@google.com

**Inderjit Dhillon**
Google & Department of Computer Science
University of Texas at Austin
isd@google.com

**Prateek Jain**
Google Research
Bengaluru, India, 560016
prajain@google.com

## Abstract

We study the problem of clustering graphs with additional side-information of node features. The problem is extensively studied, and several existing methods exploit Graph Neural Networks to learn node representations [29]. However, most of the existing methods focus on generic representations instead of their cluster-ability or do not scale to large scale graph datasets. In this work, we propose S³GC which uses contrastive learning along with Graph Neural Networks and node features to learn clusterable features. We empirically demonstrate that S³GC is able to learn the correct cluster structure even when graph information or node features are individually not informative enough to learn correct clusters. Finally, using extensive evaluation on a variety of benchmarks, we demonstrate that S³GC is able to significantly outperform state-of-the-art methods in terms of clustering accuracy – with as much as 5% gain in NMI – while being scalable to graphs of size 100M.

## 1 Introduction

Graphs are commonplace data structures to store information about entities/users, and have been investigated for decades [5, 15, 54, 31, 8, 57]. In modern ML systems, the entities/nodes are often equipped with *vector embeddings* from different sources. For example, authors are nodes in a citation graph and can be equipped with embeddings of the title/content of the authored papers [16, 41] as relevant side information. Owing to the utility of graphs in large-scale systems, tremendous progress has been made in the domain of supervised learning from graphs and node features, with Graph Neural Networks (GNNs) headlining the state-of-the-art methods [28, 19, 52]. However, typical real-world ML workflows start with unsupervised data analysis to better understand the data and design supervised methods accordingly. In fact, many times clustering is a key tool to ensure scalability to web-scale data [26]. Furthermore, even independent of supervised learning, clustering the graph data with node features is critical for a variety of real-world applications like recommendation, routing, triaging [6, 2, 32] etc.

Effective graph clustering methods should be scalable, especially with respect to the number of nodes, which can be in millions even for a moderate-scale system[57]. Furthermore, in the presence of side-information, the system should be able to use both the *views* – node features and graph information – of the data "effectively". For example, the method should be more accurate than *single-view* methods that either consider only the graph information [27] or only the node feature

---

[*]work done while the author was an intern at Google Research
[†]Now at University of Illinois, Urbana-Champaign

36th Conference on Neural Information Processing Systems (NeurIPS 2022).

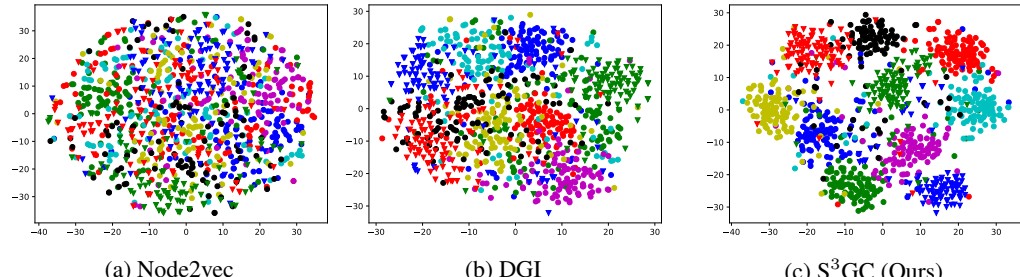

|  (a) Node2vec | (b) DGI | (c) S³GC (Ours) |

Figure 1: tSNE visualization of embeddings when applied to the data model given in Section 3.5. SBM parameters $p,q$ are such $p = q + .18$, while $\sigma_c = \sigma - .1$, i.e., both graph information and feature information are separately insufficient for clustering (see Table 1). S³GC is able to well-separate all the clusters while Node2vec [18] and DGI [53] have a significant amount of cluster overlap.

information [33, 43, 7]. This problem of graph clustering with side information has been extensively studied in the literature [61]; see Section 2 for a review of the existing and recent methods. Most methods map the problem to that of learning vector embeddings and then apply standard k-means [33] style clustering techniques. However, such methods – like Node2vec [18] – don't explicitly optimize for clusterability, therefore the resulting embeddings might not be suitable for effective clustering. Furthermore, several existing methods tend to be highly reliant on the graph information and thus tend to perform poorly when graph information is noisy/incomplete. Finally, several existing methods such as GraphCL [58] propose expensive augmentation and training modules, and thus do not scale to realistic web-scale datasets.

We propose S³GC which uses a one-layer GNN encoder to combine both the graph and node-feature information, along with graph only and node feature only encodings. S³GC applies contrastive learning to ensure that the embedding of a node is close to "near-by" nodes – obtained by random walk – while being far away from all other nodes. That is, S³GC explicitly addresses the above three mentioned challenges: a) S³GC is based on contrastive learning which is known to promote linear separability and hence clustering [20], b) S³GC carefully combines information from both the graph view and the feature view, thus performs well when one of the views is highly noisy/incomplete, c) S³GC use a light-weight encoder and simple random walk based sampler/augmentation, and can be scaled to hundreds of millions of nodes on a single virtual machine (VM).

For example, consider a dataset where the adjacency matrix of the graph is sampled from a stochastic block model with 10 clusters; let probability of an edge between nodes from same cluster is $p$ and from different clusters is $q$. Furthermore, features of each node are also sampled from a mixture of 10 Gaussians where $\sigma_c$ is the distance between any two cluster centers while $\sigma$ is the standard deviation of each Gaussian. Now, consider a setting where $p > q$ but $p, q$ are close, hence information from the graph structure is *weak*. Similarly, $\sigma_c < \sigma$ but they are *close*. Figure 1 plots two-dimensional tSNE projection [51] of embeddings learned by the state-of-the-art Node2vec[18] and DGI[53] methods, along with S³GC. Note that while Node2vec's objective function is optimized well, the embeddings do not appear to be separable. DGI's embeddings are better separated, still there is a significant overlap. In contrast, S³GC is able to produce well-separated embeddings due to the contrastive learning objective along with explicit utilization of both data views.

We conduct extensive empirical evaluation of S³GC and compare it to a variety of baselines and standard state-of-the-art benchmarks, particularly: Spectral Clustering[43], k-means[33], METIS[27], Node2vec[18], DGI[53], GRACE[62], MVGRL[21] and BGRL[48]. Overall, we observe that our method consistently outperforms Node2vec, DGI – SOTA scalable methods – on all seven datasets, achieving as much as 5% higher NMI than both the methods. For two small scale datasets, our method is competitive with MVGRL method, but MVGRL does not scale to even moderate sized datasets with about 2.5M nodes and 61M edges, while our method scales to datasets with 111M nodes and 1.6B edges.

## 2 Related Work

Below, we discuss works related to various aspects of graph clustering and self-supervised learning, and place our contribution in the context of these related works.

**Graph OR features-only clustering:** Graph clustering is a well-studied problem, and several techniques address the problem including Spectral Clustering **(SC)** [43], Graclus [12], **METIS** [27], **Node2vec** [18], and DeepWalk [40]. In particular, Node2Vec [18] is a probabilistic framework that is an extension to DeepWalk, and maps nodes to low-dimensional feature spaces such that the likelihood of preserving the local and global neighborhood of the nodes is maximized. In the setting of node-features only data, **k-means** clustering is one of the classical methods, in addition to several others like agglomerative clustering [44], density based clustering [59], and deep clustering [7].

As demonstrated in Figure 1 and Table 1, $S^3GC$ attempts to exploit both the views, and if both views are meaningful then it can be significantly more accurate than single-view methods.

**Self Supervised Learning:** Self-supervised learning methods have demonstrated that they can learn linearly separable features/representations in the absence of any labeled information. Typical approach is to define instance-wise "augmentations" and then pose the problem as that of learning contrastive representations that map instance augmentations close to the instance embedding, while pushing it far apart from all other instance embeddings. Popular examples include MoCo [22], MoCo v2 [11], **SimCLR** [9], and BYOL [17]. Such methods require augmentations, and as such do not apply directly to the graph+node-features clustering problem. $S^3GC$ uses simple random walk based augmentations to enable contrastive learning based techniques.

**Graph Clustering with Node Features:** To exploit both the graph and feature information, several existing works use the approach of autoencoder. That is, they encode nodes using Graph Neural Networks (**GNN**) [28], with the goal that inner-product of encodings can reconstruct the graph structure; **GAE** and **VGAE** [29] use this technique. GALA [38], ARGA and ARVGA [37] extend the idea by using Laplacian Sharpening and generative adversarial learning. Structural Deep Clustering Network (SDCN) [4] jointly learns an Auto-Encoder (AE) along with a Graph Auto-Encoder (GAE) for better node representations, while Deep Fusion Clustering Network (DFCN) [50] merges the representations learned by AE and GAE for consensus representation learning. Since AE type approaches attempt to solve a much harder problem, their accuracy in practice lags significantly to the state-of-the-art; for example, see Table 3 in [21] which shows that such techniques can be 5-8% less accurate. **MinCutPool** [42] and **DMoN** [49] extend spectral clustering with graph encoders, but the resulting problem is somewhat unstable and leads to relatively poor partitions; see Table 3.

**Graph Contrastive Learning:** Recently several papers have explored contrastive Graph Representation Learning based approaches and have demonstrated state-of-the-art performance. Deep Graph Infomax (**DGI**) [53] is based on MINE [24] method, and is one of the most scalable method with nearly SOTA performance. It uses edge permutations to learn augmentations and embeddings. Infograph [47] extends the DGI idea to learn unsupervised representations for graphs as well. GraphCL [58] design a framework with four types of graph augmentations for learning unsupervised representations of graph data using a contrastive objective. **MVGRL** [21] extends these ideas by performing node diffusion and contrasting node representations with augmented graph representations while **GRACE** [62] maximizes agreement of node embeddings across two corrupted views of the graph. Bootstrapped Graph Latents (**BGRL**) [48] adapts the BYOL [17] methodology to the graph domain, and eliminates the need for negative sampling by minimizing an invariance based loss for augmented graphs within a batch. While these methods are able to obtain more powerful embeddings, the augmentations and objective function setup become expensive, and hence they are hard to scale to large datasets beyond $\sim$ 1M nodes. In contrast, $S^3GC$ is able to provide competitive or better clustering accuracy, while still being scalable to graphs of size 100M nodes.

## 3 $S^3GC$: Scalable Self-Supervised Graph Contrastive Clustering

In this section, we first formally introduce the problem of graph clustering and notations. Then we discuss challenges faced by the current methods and outline the framework of our method $S^3GC$. Finally, we detail each component of our method and highlight the overall training methodology.

### 3.1 Problem Statement and Notations

Consider a graph $G = (V, E)$ with the vertex set $V = \{v_1, \cdots, v_n\}$ and the edge set $E \subseteq V \times V$, where $|E| = m$. Let $A \in \mathbb{R}^{n \times n}$ be the adjacency matrix of $G$, where $A_{ij} = 1$ if $(v_i, v_j) \in E$, else $A_{ij} = 0$. Let $X \in \mathbb{R}^{n \times d}$ be the node attributes or feature matrix, where the $i$-th row $X_i$ denotes the $d$-dimensional feature vector of node $i$. Given the graph $G$ and attributes $X$, the aim is to partition the graph $G$ into $k$ partitions $\{G_1, G_2, G_3, ..., G_k\}$ such that nodes in the same cluster are similar/close to each other in terms of the graph structure as well as in terms of attributes.

Now, in general, one can define several loss functions to evaluate quality of clustering but that might not reflect the underlying ground truth. So, to evaluate the quality of clustering, we use standard benchmarks which have ground truth labels apriori. Furthermore, Normalized Mutual Information (NMI) between the ground truth labels and the estimated cluster labels is used as the key metric. NMI between two labellings $Y_1$ and $Y_2$ is defined as:

$$\text{NMI}(Y_1, Y_2) = \frac{2 \cdot I(Y_1, Y_2)}{H(Y_1) + H(Y_2)} \tag{1}$$

where $I(Y_1, Y_2)$ is the Mutual Information between labellings $Y_1$ and $Y_2$, and $H(\cdot)$ is the entropy. Normalized Adjacency Matrix is denoted by $\tilde{\mathbf{A}} = \mathbf{D}^{-\frac{1}{2}} \mathbf{A} \mathbf{D}^{-\frac{1}{2}} \in \mathbb{R}^{n \times n}$ where $\mathbf{D} = diag(\mathbf{A1}_N)$ is the degree matrix. We also compute a $k-$hop Diffusion Matrix, denoted by $\mathbf{S_k} = \sum_{i=0}^{k} \alpha_i \tilde{\mathbf{A}}^i \in \mathbb{R}^{n \times n}$, where $\alpha_i \in [0, 1] \ \forall i \in [k]$, and $\sum_i \alpha_i \leq 1$. Intuitively, $k-$hop diffusion matrix captures a weighted average of $k$-hop neighbourhood around every node. For specific $\alpha_i$ and for $k = \infty$, diffusion matrix can be computed in closed form [30, 36]. However, in this work we focus on finite $k$.

## 3.2  Challenges in Graph Clustering

Clustering in general is a challenging problem as the underlying function to evaluate quality of the clustering solution is unknown apriori. However, graph partitioning/clustering with attributes poses several more challenges. In particular, scaling the methods is challenging as graphs are sparse data structures, while neural network based approaches produce dense artifacts. Furthermore, it is challenging to effectively combine information from the two data views: graph and the feature attributes. Node2vec [18] uses only graph structure information, DGI [53] and related methods [21, 39] are highly dependent upon attribute quality. Motivated by the above mentioned challenges, we propose S$^3$GC which uses a self-supervised variant of GNNs.

## 3.3  S$^3$GC: Scalable Self Supervised Graph Clustering – Methodology

At a high level, S$^3$GC uses a Graph Convolution Network (GCN) based encoder and optimizes it using a contrastive loss where the nodes are sampled via a random walk. Below we describe the three components of S$^3$GC and then provide the resulting training algorithm.

**Graph Convolutional Encoder:** We use a 1-layer Graph Convolutional Network [28] to encode the graph and feature information for each node:

$$\overline{\mathbf{X}} = \left( PReLU(\tilde{\mathbf{A}} \mathbf{X} \Theta) + PReLU(\mathbf{S_k} \mathbf{X} \Theta') + \mathcal{I} \right) \tag{2}$$

where $\overline{\mathbf{X}} \in \mathbb{R}^{n \times d}$ stores the learned $d$-dimensional representation of each node. Recall that $\tilde{\mathbf{A}}$ is the normalized adjacency matrix and $\mathbf{S_k}$ is the $k$-hop diffusion matrix. $\mathcal{I} \in \mathbb{R}^{n \times d}$ is a learnable matrix. $\{\Theta, \Theta'\}$ are the weights of the GCN layer, and $PReLU$ is the parameteric ReLU activation function [23]:

$$f(z_i) = z_i \ \text{if} \ z_i \geq 0, \ f(z_i) = a \cdot z_i \ \text{otherwise}, \tag{3}$$

where $a$ is a learnable parameter. Our choice of encoder makes the method scalable as a 1-layer GCN requires storing only the learnable parameters in the GPU/memory, which is small ( $O(d^2)$, where $d$ is the dimensionality of the node attributes). The parameter $\mathcal{I}$ scales only linearly with the number of nodes $n$. More importantly, we use mini-batches that reduce the memory requirement of forward and backward pass to order $O(rsd + d^2)$ where $r$ is the batch size in consideration and $s$ is the average degree of nodes, therefore making our method scalable to graphs of very large sizes as well. We provide further discussion on memory requirement of our method in Section 3.4.

**Random Walk Sampler:** Next, inspired by [40, 18], we utilise biased second order Random Walks with restarts to generate points *similar* to a given node and thus capture the local neighborhood of each node. Formally following [18], we start with a source node $u$, and simulate a random walk of length $l$. We use $c_i$ to denote the $i$-th node in the random walk starting from $c_0 = u$. Every other node in walk $c_i$ is generated from the distribution:

$$P\left(c_i = x \mid c_{i-1} = v\right) = \frac{\pi_{vx}}{Z} \text{, if } (v, x) \in E, \ \ P\left(c_i = x \mid c_{i-1} = v\right) = 0 \text{ otherwise} \tag{4}$$

where $\pi_{vx}$ is the unnormalized transition probability between nodes $v$ and $x$ and $Z$ is the normalization constant. To bias the random walks and compute the next edge $x$ we follow a methodology similar

to [18], and from node $v$ after traveling $(t, v)$, the transition probability $\pi_{vx}$ is set to $\alpha_{pq}(t, x) \cdot w_{vx}$ where $w_{vx}$ is the weight on the edge between $v$ and $x$, and the bias parameter $\alpha$ is defined by:

$$\alpha_{pq}(t, x) = \frac{1}{p}, \text{ if } d_{tx} = 0, \quad \alpha_{pq}(t, x) = 1 \text{ ,if } d_{tx} = 1, \quad alpha_{pq}(t, x) = \frac{1}{q} \text{ ,if } d_{tx} = 2, \quad (5)$$

where $p$ is the *return* parameter, controlling the likelihood of immediately revisiting a node, $q$ is the *in-out* parameter [18], allowing the search to differentiate between "inward" and "outward" nodes, and $d_{tx}$ denotes the shortest path distance between nodes $t$ and $x$. We note that $d_{tx}$ from node $t$ to $x$ can only take values $\in \{0, 1, 2\}$. Setting $p$ to a high value $(> max(q, 1))$ ensures a lesser likelihood of revisiting a node and setting it to a low value $(< min(q, 1))$ would make the walk more "local". Similarly, setting $q > 1$ would bias the random walk to nodes near $t$ and obtain a local view of the graph encouraging BFS-like behaviour, whereas a $q < 1$ would bias the walk towards nodes further away from $t$ and encourage DFS-like behaviour.

**Contrastive Loss Formulation:** Now to learn the encoder parameters, we use SimCLR style loss function where nodes generated from the random walk are considered to be *positives* while rest of the samples are considered to be negative. That is, we use graph neighborhood information to produce *augmentations* of a node. Formally, let $\mathcal{C}(u)^+ = \{c_0, c_1, ..., c_l\}$ be the nodes generated by a random walk starting at $c_0 = u$. Then, $\mathcal{C}(u)$ is the set of positive samples $p_u^+$, while the set of negatives $p_u^-$ is generated by sampling $l$ nodes from the remaining set of nodes $[n] \backslash p_u^+$. Given $p_u^+$ and $p_u^-$, we can now define the loss for each $u$ as:

$$\mathcal{L}_{SimCLR}(u) = -\frac{\sum_{v \in p_u^+} exp(sim(\overline{\mathbf{X}}_u, \overline{\mathbf{X}}_v))}{\sum_{v \in p_u^+} exp(sim(\overline{\mathbf{X}}_u, \overline{\mathbf{X}}_v)) + \sum_{v' \in p_u^-} exp(sim(\overline{\mathbf{X}}_u, \overline{\mathbf{X}}_{v'}))} \quad (6)$$

where $sim$ is some similarity function, for example inner product: $sim(u, v) = \frac{u^T v}{\|u\|\|v\|}$.

Note that SimCLR style loss functions have been shown to lead to "linearly separable" representations [20] and hence aligns well with the clustering objective [55, 10]. In contrast, loss functions like those used in Node2vec [18] might not necessarily lead to "clusterable" representations which is also indicated by their performance on synthetic as well as real-world datasets.

### 3.4 Algorithm

Now that we have discussed the individual components of our method, we describe the overall training methodology in Algorithm 1. We begin with the initialization of the learnable parameters in line 1. In line 4,5 we generate the positive and negative samples for each node in the current batch. Since we operate with embeddings of only the nodes in batch and their positive/negative samples, we take a union of these to create a "node set" in line 6. This helps in reducing the memory requirements of our algorithm, since we do not do forward/backward pass on the entire $AX$, but only on the nodes needed for the current batch. Once we have the node set, we compute representations for the nodes in the current batch using a forward pass in line 8, compute the loss for nodes in this set in line 9, and perform back-propagation to generate the gradient updates for the learnable parameters in line 10. Finally, we update the learnable parameters in line 11 and repeat the process for the next batch.

**Space Complexity:** The space complexity for the forward and backward pass of our algorithm is $O(rsd + d^2)$, where $r$ is the batch size, $s$ is the average degree of nodes, and $d$ is the attribute dimension. The process of random walk generation is fast and can be done in memory, which is abundantly available and highly parallelizable. Therefore, storing the graph structure in memory for sampling of positives doesn't create a memory bottleneck and takes $O(m)$ space. For all the datasets other than ogbn-papers100M, we store the $AX, SX$, and $\mathcal{I}$ in the GPU memory as well, requiring additional $O(nd)$ space. However, for very large-scale datasets, one can conveniently store these in the memory itself and interface with the GPU when required, thereby restricting the GPU memory requirement to $O(rsd + d^2)$.

**Time Complexity:** The forward and backward computation for a given batch takes $O(rsd^2)$ time. Hence, for $n$ nodes, batch size of $r$, and $K$ epochs, time complexity is $O(Knsd^2)$.

**Embedding property:** Detecting communities ideally requires nodes to be clustered based on their position, rather than structural similarities. We show in Appendix C that S$^3$GC produces positional embeddings [46].

**Code:** Implementation code of S$^3$GC is available at: https://github.com/devvrit/S3GC

**Algorithm 1** S$^3$GC: Training and Backpropagation

---

**Input:** Graph **G**, Matrices $\tilde{\mathbf{A}}\mathbf{X} \in \mathbb{R}^{n \times d}$ and $\mathbf{S_k}\mathbf{X} \in \mathbb{R}^{n \times d}$, number of epochs **K**, Batched inputs of nodes $B$, Self-supervised Loss formulation $\mathcal{L}_{SimCLR}$, Encoder definition ENC, Learning Rate $\eta$

1:  Initialize model parameters: $\boldsymbol{\Theta}, \boldsymbol{\Theta}', \mathcal{I}$
2:  **for** epoch = $1, 2, \ldots K$ **do**
3:     **for** each batch: $b \in B$ **do**
4:         Generate Positive Samples $p_v^+$ using biased random walks Section 3.3 $\forall\, v \in b$
5:         Generate Negative Samples $p_v^-$ using random sampling $\forall\, v \in b$
6:         Compute the node set $N_b$ : $\mathsf{UNION}(p_v^+, p_v^-)\, \forall\, v$
7:         Select subset rows $(\tilde{\mathbf{A}}\mathbf{X})_{N_b}$ and $(\mathbf{S_k}\mathbf{X})_{N_b}$ corresponding to the node set $N_b$
8:         Forward Pass to compute the representations: $\overline{\mathbf{X}} \leftarrow \mathsf{ENC}((\tilde{\mathbf{A}}\mathbf{X})_{N_b}, (\mathbf{S_k}\mathbf{X})_{N_b}, \boldsymbol{\Theta}, \boldsymbol{\Theta}', \mathcal{I})$
9:         Compute loss using the self-supervised formulation: $\mathcal{L}(\overline{\mathbf{X}})$
10:        Compute Gradients for learnable parameters at time $t$: $\mathbf{u_t}(\boldsymbol{\Theta}, \boldsymbol{\Theta}', \mathcal{I}) \leftarrow \nabla_{\boldsymbol{\Theta}, \boldsymbol{\Theta}', \mathcal{I}} \mathcal{L}(\overline{\mathbf{X}})$
11:        Refresh the parameters: $(\boldsymbol{\Theta}, \boldsymbol{\Theta}', \mathcal{I})_{t+1} \leftarrow (\boldsymbol{\Theta}, \boldsymbol{\Theta}', \mathcal{I})_t - \frac{\eta}{|b|}\mathbf{u_t}(\boldsymbol{\Theta}, \boldsymbol{\Theta}', \mathcal{I})$

**Output:** $\overline{\mathbf{X}}; \boldsymbol{\Theta}, \boldsymbol{\Theta}', \mathcal{I}$

---

Table 1: **Results on experiments using dataset generated from Stochastic Block Models**. SC represents Spectral Clustering [43] on the graph, k-means utilises only the attributes, Node2vec[18] uses the graph structure and DGI [53] utilizes both. We experiment with two variants of our method S$^3$GC-$\mathcal{I}$ using only $\mathcal{I}$ as the learnable embeddings without using the attributes and S$^3$GC using both graph and attributes, and evaluate the quality of clustering using mean NMI, reported over 3 runs.

| SBM Parameters | | | | Baselines: NMI | | | | Our Method: NMI | |
|---|---|---|---|---|---|---|---|---|---|
| p | q | $\sigma_c$ | $\sigma$ | SC | k-means | Node2Vec | DGI | S$^3$GC$-\mathcal{I}$ | S$^3$GC |
| 0.65 | 0.40 | 0.5 | 1.5 | 0.96 | 0.25 | 0.96 | 0.85 | **0.99** | **1.00** |
| 0.63 | 0.45 | 0.5 | 1.5 | 0.45 | 0.25 | 0.47 | 0.55 | **0.64** | **0.71** |
| 0.63 | 0.45 | 1.0 | 1.5 | 0.45 | 0.90 | 0.47 | 0.86 | **0.64** | **0.91** |

### 3.5 Synthetic Dataset – Stochastic Blockmodel with Gaussian Features

To better understand the working of our method in scenarios with varied quality of the graph structure and node attributes, we propose a study on a synthetic dataset using Stochastic Block Models (SBM)[1] with Gaussian features. For a given parameter $k$, the SBM [45] constructs a graph $G = (V, E)$ with $k$ partitions of nodes $V$. The probability of an intra-cluster edge is $p$ and an inter-cluster edge is $q$, where $p > q$.[3] Similar studies have been proposed for benchmarking of GNNs [13] and Graph clustering methods[14, 49] using SBM. In this work, we create an attributed SBM model, where each node has an $s$-dimensional attribute associated with it. Following the setup in [49], for $k$ clusters (partitions) we generate $k$ cluster centers using $s$-multivariate normal distributions $\mathcal{N}(0_s, \sigma_c^2 \cdot I_s)$, where $\sigma_c^2$ is a hyperparameter we define. Then attributes of nodes of a given cluster are sampled from an $s$-multivariate gaussian distribution with the corresponding cluster center and $\sigma^2 I$ variance. The ratio $\sigma_c^2/\sigma^2$ controls the expected value of the classical *between* vs *within* sum-of-squares of the clusters.

We compare our method with: k-means on the attributes, Spectral Clustering [43], DGI [53], and Node2vec [18]. This choice of baseline methods focuses on different facets of graph data and clustering across which we want to assess the performance of our method. k-means on attributes utilizes only the nodes attribute information. Spectral Clustering is a non-trainable classical algorithm commonly used for solving SBMs, but uses only the graph-structure information. Similarly, Node2vec is a common graph-embedding trainable algorithm that utilizes only the structural information. DGI is a scalable SOTA self-supervised graph representation learning algorithm that uses both structure as well as node attributes.

To demonstrate the effectiveness of our choice of loss formulation, we also run our method without using any attribute information and using only the learnable embedding $\mathcal{I} \in \mathbb{R}^{n \times d}$, i.e. $\overline{\mathbf{X}} = \mathcal{I}$.

---

[3]Note that these are parameters for the SBM dataset generation, unrelated to the random walk sampling parameters in the S$^3$GC model.

Table 2: **Datasets and Setup.**

| Scale | Dataset | # nodes | # edges | feat dim | # classes |
|---|---|---|---|---|---|
| **Small Scale** | Cora | 2,708 | 5,278 | 1,433 | 7 |
| | Citeseer | 3,327 | 4,614 | 3,703 | 6 |
| | Pubmed | 19,717 | 44,325 | 500 | 3 |
| **Moderate/Large Scale** | ogbn-arxiv | 169,343 | 1,166,243 | 128 | 40 |
| | Reddit | 232,965 | 23,213,838 | 602 | 41 |
| | ogbn-products | 2,449,029 | 61,859,140 | 100 | 47 |
| **Extra-Large Scale** | ogbn-papers100M | 111,059,956 | 1,615,685,872 | 128 | 172 |

**Setup and Observations:** We set the number of nodes $n = 1000$ and number of clusters $k = 10$, where each cluster contains $n/k = 100$ nodes, and vary $p$ and $q$ to generate graphs of different structural qualities. Varying $\sigma_c^2/\sigma^2$ controls the quality of the attributes. The first row in table 1 represents a graph with high structural as well as attribute quality. The second row represents low structural as well as low attribute quality. While the last row represents low structural but high attribute quality. We make several observations: 1) Even without using any attribute information, our method performs significantly better as compared to other structure-only based methods like Spectral Clustering and Node2Vec, which demonstrates the effectiveness of our loss formulation and training methodology that promotes clusterability, which is also in line with recent observations [10, 55]. 2) We observe that DGI depends highly on the quality of the attributes and is not able to utilize the high-quality graph structure as well, when the attributes are noisy. In contrast, our method uses both sources of information effectively and performs reasonably well even when even only one of the structure or attribute quality is high (first and the last row in the table).

**Visualization of the Embeddings:** We further observe the quality of the generated embeddings using t-SNE[51] projected in 2-dimensions. Figure 1 corresponds to the second setting with weak graph and weak attributes, where we observe that $S^3GC$ generates representations which are more *cluster-like* as compared to the other methods. Additionally, we note that $S^3GC$ shows similar behaviour in the other two settings as well, the plots for which are provided in the Appendix.

## 4 Empirical Evaluation

We conduct extensive experiments on several node classification benchmark datasets to evaluate the performance of $S^3GC$ as compared to key state-of-the-art (SOTA) baselines across multiple facets associated with Graph Clustering.

### 4.1 Datasets and Setup

**Datasets:** We use 3 small scale, 3 moderate/large scale, and 1 extra large scale dataset from GCN [28], GraphSAGE [19] and the OGB-suite [25] to demonstrate the efficacy of our method. The details of the datasets are given in Table 2 and additional details of the sources are mentioned in Appendix.

**Baselines:** We compare our method with k-means on features and 8 recent state-of-the-art baseline algorithms, including MinCutPool [3], METIS [27], Node2vec [18], DGI [53], DMoN [49], GRACE [62], BGRL [48] and MVGRL [21]. We choose baseline methods from a broad spectrum of methodologies, namely methods that utilize only the graph structure, methods that utilize only the features and specific methods that utilize a combination of the graph structure and attribute information to provide an exhaustive comparison across important facets of graph learning and clustering. METIS [27] is a well-known and scalable classical method for graph partitioning using only the structural information. Similarly, Node2vec[18] is another scalable graph embedding technique that utilizes random walks on the graph structure. MinCutPool[3] and DMoN [49] are graph clustering techniques motivated by the normalized MinCut objective [42] and Modularity [35] respectively. DGI is a SOTA self-supervised method utilizing both graph structure and features, that motivated a line of work [21, 39] based on entropy maximization between local and global views of a graph. GRACE[62], in contrast to DGI's methdology, contrasts embeddings at the node level itself, by forming two views of the graph and maximizing the embedding of the same nodes in the two views. BGRL[48] and MVGRL [21] are recent SOTA methods for performing self-supervised graph representation learning.

**Metrics:** We measure 5 metrics which are relevant for evaluating the quality of the cluster asssignments following the evaluation setup of [56, 21]: Accuracy, Normalized Mutual Information (NMI), Completeness Score (CS), Macro-F1 Score (F1), and Adjusted Rand Index (ARI). For all these aforementioned metrics, a higher value indicates better clustering performance. We generate the

representations using each representation-learning method and then perform k-means clustering on the embeddings to generate the cluster assignments used for evalution of these metrics.

**Detailed Setup.** We consider the unsupervised learning setting for all the seven datasets where the graph and features corresponding to all the datasets are available. We use the labels only for evaluating the quality of the cluster assignments generated by each method. For the baselines, we use the official implementations provided by the authors without any modifications. All experiments are repeated 3 times and the mean values are reported in the Table 3. We highlight the highest value as well as any other values within 1 standard deviation of the mean of the best performing method, and report the results with standard deviations in the Appendix, due to space constraints. We utilize a single Nvidia A100 GPU with 40GB memory for training each method for a maximum duration of 1 hour for each experiment in Table 3. For ogbn-papers100M we allow upto $\sim 24$ hours of training and upto $300$GB main memory in addition. We provide a mini-batched and highly scalable implementation of our method S$^3$GC in PyTorch such that experiments on all datasets other than ogbn-papers100M easily fit in the aforementioned GPU. For the ogbn-papers100M dataset, the *forward* and *backward* pass in S$^3$GC are performed in the GPU, with an interfacing with the CPU memory to store the graph, $AX$, and $SX$, and to maintain and update $\mathcal{I}$, with minimal overheads. We also provide a comparison of the time and space complexity for each method in the Appendix.

**Hyperparameter Tuning:** S$^3$GC requires selection of minimal hyperparameters: we use $k = 2$ for the k-hop Diffusion Matrix $S_k$ which offers the following advantages: 1) $S_2X = \alpha_0 X + \alpha_1 \tilde{\mathbf{A}}X + \alpha_2 \tilde{\mathbf{A}}^2 X$ is a finite computation which can be pre-computed and only requires 2 sparse-dense matrix multiplications. 2) We chose $\alpha_0 > \alpha_1 > \alpha_2$, giving a higher weight to 0-hop neighbourhood attributes $X$ which allows S$^3$GC to exploit the rich information from good quality attributes even when the structural information is not very informative. 3) Two-hop neighbourhood intuitively captures all the features of nodes with similar attributes while maintaining scalability. This is motivated by the 2-hop and 3-hop choice of neighborhoods in [19] and [25] for these datasets. We additionally tune the learning rate, batch size and random walk parameters, namely the walk length $l$ while using the default values of $p = 1$ and $q = 1$ for the bias parameters in the walk. We perform model selection based on the NMI on the validation set and evaluate all the metrics for this model. Additional details regarding the hyperparameters are mentioned in the Appendix due to space restrictions.

### 4.2 Results

Table 3 compares clustering performance of S$^3$GC to a number of baseline methods on datasets of three different scales. For the small scale datasets, namely Cora, Citeseer and Pubmed, we observe that MVGRL outperforms all methods. We also note that MVGRL's performance in our experiments, using the author's official implementation with extensive hyperparameter tuning is slightly lower than the reported values, as has been reported by other works as well [60]. Nonetheless, we use these values for comparison and observe that S$^3$GC also performs either competitively or is slightly inferior to MVGRL's accuracy. For example, on the Cora dataset, S$^3$GC is within $\sim 2\%$ of MVGRL's performance and outperforms all the other baseline methods, while on the Pubmed dataset, S$^3$GC is within $\sim 1.5\%$ of MVGRL's performance. Next, we observe the performance on moderate/large scale datasets and note that S$^3$GC significantly outperforms baselines such as k-means, MinCutPool, METIS, Node2vec, DGI and DMoN. Notably, S$^3$GC is $\sim 5\%$ better on ogbn-arxiv, $\sim 1.5\%$ better on Reddit and $\sim 4\%$ better on ogbn-products in terms of clustering NMI as compared to the next best method. The official implementations of GRACE, BGRL, and MVGRL do not scale to datasets with >200k nodes, running into Out of Memory (OOM) errors due to the non-scalable implementations, sub-optimal memory utilization, or the non-scalable methodology. For example, MVGRL proposes diffusion matrix as the alternate view of graph structure, which is a dense $n \times n$ matrix - hence, not scalable.

We also note that S$^3$GC performs reasonably well in settings where the node attributes are not very informative while the graph structure is useful, as evident from the performance on the Reddit dataset. k-means on the node attributes gives an NMI of only $\sim 10\%$ while methods like METIS and Node2vec perform well using the graph structure. Methods like DGI which depend heavily on the quality of the attributes, thus suffer a degradation in performance having a clustering NMI of only $\sim 30\%$, while S$^3$GC which uses both the attributes and graph information effectively outperforms all the other methods and generates clustering with an NMI of $\sim 80\%$.

**ogbn-papers100M**: Finally, we compare the performance of S$^3$GC on the extra-large scale dataset with 111M nodes and 1.6B edges in Table 4, and note that only k-means, Node2vec and DGI scale

Table 3: **Comparison of clustering obtained by our method S$^3$GC to several state-of-the-art methods..** Metrics for evaluation across different datasets and experiments are Accuracy, NMI, CS, F1 and ARI as described in Section 4. We use the official implementations provided by the authors for all the methods and provide additional details in the Appendix. * denotes that the method ran Out of Memory (OOM) while trying to run the experiments on the hardware as specified in Section 4. ‖ indicates that the method did not converge.

| Dataset | Metric | Baseline | | | | | | | | | Ours |
|---|---|---|---|---|---|---|---|---|---|---|---|
| | | k-means | MinCutPool | METIS | Node2vec | DGI | DMoN | GRACE | BGRL | MVGRL | S$^3$GC |
| Cora | Accuracy | 0.350 | 0.490 | 0.540 | 0.612 | 0.726 | 0.517 | 0.739 | 0.742 | **0.763** | 0.742 |
| | NMI | 0.173 | 0.410 | 0.396 | 0.444 | 0.571 | 0.473 | 0.570 | 0.584 | **0.608** | 0.588 |
| | CS | 0.171 | 0.407 | 0.384 | 0.449 | 0.568 | 0.406 | 0.562 | 0.595 | **0.617** | 0.586 |
| | F1 | 0.360 | 0.471 | 0.518 | 0.621 | 0.692 | 0.574 | **0.725** | 0.691 | 0.716 | **0.721** |
| | ARI | 0.127 | 0.317 | 0.308 | 0.329 | 0.511 | 0.301 | 0.527 | 0.534 | **0.566** | 0.544 |
| Citeseer | Accuracy | 0.421 | 0.537 | 0.413 | 0.421 | 0.686 | 0.385 | 0.631 | 0.675 | **0.703** | 0.688 |
| | NMI | 0.199 | 0.295 | 0.17 | 0.240 | 0.435 | 0.303 | 0.399 | 0.422 | **0.459** | 0.441 |
| | CS | 0.205 | 0.296 | 0.167 | 0.264 | 0.436 | 0.251 | 0.398 | 0.423 | **0.460** | 0.441 |
| | F1 | 0.394 | 0.516 | 0.400 | 0.401 | 0.643 | 0.437 | 0.603 | 0.631 | **0.654** | 0.643 |
| | ARI | 0.142 | 0.262 | 0.150 | 0.116 | 0.445 | 0.200 | 0.377 | 0.428 | **0.471** | 0.448 |
| Pubmed | Accuracy | 0.601 | 0.521 | 0.693 | 0.641 | 0.657 | 0.351 | 0.637 | 0.654 | 0.675 | **0.713** |
| | NMI | 0.314 | 0.214 | 0.297 | 0.288 | 0.322 | 0.257 | 0.308 | 0.315 | **0.345** | 0.333 |
| | CS | 0.344 | 0.247 | 0.291 | 0.288 | 0.332 | 0.179 | 0.321 | 0.325 | **0.355** | 0.337 |
| | F1 | 0.592 | 0.445 | 0.682 | 0.634 | 0.654 | 0.343 | 0.628 | 0.649 | 0.672 | **0.703** |
| | ARI | 0.281 | 0.175 | 0.323 | 0.258 | 0.292 | 0.108 | 0.276 | 0.285 | 0.310 | **0.345** |
| ogbn-arxiv | Accuracy | 0.176 | 0.242 | 0.209 | 0.290 | 0.314 | 0.250 | * | 0.227 | * | **0.350** |
| | NMI | 0.216 | 0.380 | 0.345 | 0.406 | 0.412 | 0.356 | * | 0.321 | * | **0.463** |
| | CS | 0.198 | 0.344 | 0.312 | 0.370 | 0.379 | 0.326 | * | 0.293 | * | **0.420** |
| | F1 | 0.121 | 0.198 | 0.167 | 0.220 | 0.230 | 0.190 | * | 0.166 | * | **0.230** |
| | ARI | 0.074 | 0.139 | 0.126 | 0.190 | 0.223 | 0.127 | * | 0.130 | * | **0.270** |
| Reddit | Accuracy | 0.089 | ‖ | 0.524 | 0.709 | 0.224 | 0.529 | * | * | * | **0.736** |
| | NMI | 0.114 | ‖ | 0.727 | **0.792** | 0.306 | 0.628 | * | * | * | 0.807 |
| | CS | 0.112 | ‖ | 0.697 | 0.795 | 0.300 | 0.678 | * | * | * | 0.821 |
| | F1 | 0.068 | ‖ | 0.495 | **0.551** | 0.183 | 0.260 | * | * | * | 0.560 |
| | ARI | 0.029 | ‖ | 0.470 | 0.640 | 0.170 | 0.502 | * | * | * | **0.745** |
| ogbn-products | Accuracy | 0.200 | 0.257 | 0.294 | 0.357 | 0.320 | 0.304 | * | * | * | **0.402** |
| | NMI | 0.273 | 0.430 | 0.468 | 0.489 | 0.467 | 0.428 | * | * | * | **0.536** |
| | CS | 0.236 | 0.360 | 0.401 | 0.425 | 0.405 | 0.367 | * | * | * | **0.463** |
| | F1 | 0.124 | 0.180 | 0.220 | **0.247** | 0.192 | 0.210 | * | * | * | 0.250 |
| | ARI | 0.082 | 0.130 | 0.145 | 0.170 | 0.174 | 0.139 | * | * | * | **0.230** |

Table 4: **Results of comparison of the embeddings generated by our method S$^3$GC as compared to different scalable methods on ogbn-papers100M with 111M nodes and 1.6B edges**.

| Method | ogbn-papers100M | | | | |
|---|---|---|---|---|---|
| | Accuracy | NMI | CS | F1 | ARI |
| k-means | 0.144 | 0.368 | 0.342 | 0.101 | 0.074 |
| Node2vec | **0.175** | 0.380 | 0.352 | **0.099** | **0.112** |
| DGI | 0.151 | 0.416 | 0.386 | **0.111** | 0.096 |
| S$^3$GC (Ours) | **0.173** | **0.453** | **0.430** | **0.118** | **0.110** |

to this dataset size and run in a reasonable time of $\sim 24$ hours. We observe that S$^3$GC seamlessly scales to this dataset and significantly outperforms methods utilizing only the features (k-means) by $\sim 8.5\%$, only graph structure (Node2vec) by $\sim 7\%$ and both (DGI) by $\sim 4\%$ in terms of clustering NMI on the ogbn-papers100M dataset.

**Ablation Study on Hyperparameters:** We perform detailed ablation studies to investigate the stability of S$^3$GC's clustering and provide the same, in the Appendix. We find that S$^3$GC is robust to its few hyperparameters such as walk-length and batch size, enabling a near-optimal choice. We note that smaller walk lengths $\sim 5$ are an optimal choice across datasets, since they are able to include the "right" positive examples in the batch, while using larger walk lengths may degrade the performance due to the inclusion of nodes belonging to other classes in the positive samples. This helps in scalability as well, as we need to sample only a few positives per node. While small batches take more time *per-epoch* but converge faster, larger batch sizes are better in *per-epoch* training time, but require more epochs to converge. Both, however, enjoy similar performance in terms of the quality of the clustering.

## 4.3 Novelty of S$^3$GC's Design Choices

Our design choices have unique roles to play which make S$^3$GC both scalable and accurate, by effectively utilizing structure as well as node attribute information for learning clusterable representa-

Table 5: **Comparison between encoder choices.** $S^3GC_{GCN}$ refers to $S^3GC$ where we use encoder $\bar{X} = PreLU(\tilde{A}X\Theta) + PreLU(\tilde{S}_k X\Theta')$. While $S^3GC_{\mathcal{I}}$ refers to encoder $\bar{X} = \mathcal{I}$. We compare with a structure-only method Node2Vec and an attribute-dependent method DGI.

| Dataset | Node2Vec | DGI | $S^3GC_{GCN}$ | $S^3GC_{\mathcal{I}}$ | $S^3GC$ |
|---------|----------|-----|---------------|----------------------|---------|
| ogbn-arxiv | 0.406 | 0.412 | 0.460 | 0.444 | 0.463 |
| reddit | 0.792 | 0.306 | 0.777 | 0.808 | 0.807 |
| ogbn-products | 0.489 | 0.467 | 0.528 | 0.535 | 0.536 |

tions. We describe their importance and contrast with other possible design choices in this section, highlighting how these choices put together work the most effectively empirically, contributing to $S^3GC$'s novelty.

**Encoder:** Using a multilayered GCN[28] to capture local graph structure along with attribute information increases the space required to $O(nnz(A))$, making any method non-scalable for very large graphs. This issue is also faced by existing methods like MVGRL [21] which compute the entire diffusion matrix, and hence run into OOM errors on larger datasets as discussed earlier. Hence, we use a 1-layered GCN and precompute $AX$, requiring only $O(nd)$ space. Intuitively, we see that it is important to utilize both the attribute information as well as structural information in the encoder. With this motivation, we design $S^3GC$ to capture attribute information using a 1-layer GCN and capture structural information using a learnable parameter $\mathcal{I}$ (eq. (2)). We empirically verify this intuition with experiments using either just the attribute information (for example on the Reddit dataset) or only the structural information (on the ogbn-arxiv dataset) and summarize our findings in Table 5. We find that the design choice of $S^3GC$'s Encoder is optimal for effectively capturing both sources of information and removing any one source, leads to considerably suboptimal performance.

**Positive and Negative nodes sampler:** Using a random walk sampler offers several advantages - it can be computed in a scalable fashion and it samples nodes from a $k$-hop neighbourhood. We consider several intuitive sampling approaches and discuss the most intuitive and simple sampling approach here: for a given node, we consider all its $k-$hop neighbourhood nodes as positives, and $r$ randomly sampled nodes as negative. This becomes non-scalable, since calculating a $k$-hop neighbourhood for all the nodes in the graph has a significant computation cost (it is equivalent to computing non-zero elements in $A^k$). Hence, for simplicity, we experiment with $k = 1$ and note that on the ogbn-arxiv dataset, using this sampling scheme and a learnable embedding as the encoder $\bar{X} = \mathcal{I}$ gives only 0.252 NMI. This is significantly lesser than using a random walk generator for sampling and the same encoding scheme $\bar{X} = \mathcal{I}$, which gives an NMI of 0.444.

**Loss function** As we already observe in Section 3.5, using the same encoder $\bar{X} = \mathcal{I}$ but a different loss function gives rise to more clusterable embeddings when using the SimCLR loss as compared to using the Node2Vec loss. The experiments on real-world datasets also reinforce these observations, as we see in Table 5 that $S^3GC_{\mathcal{I}}$ performs better than Node2Vec on ogbn-arxiv and ogbn-products datasets - where both of the methods use the same encoder and random sampler but different loss functions.

## 5 Discussion and Future Work

We introduced $S^3GC$, a new method for scalable graph clustering with node feature side-information. $S^3GC$ is a simple method based on contrastive learning along with a careful encoding of graph and node features, but it is an effective approach across all scales of data. In particular, we showed that $S^3GC$ is able to scale to graphs with 100M nodes while still ensuring SOTA clustering performance.

**Limitations and Future Work**: $S^3GC$ demonstrates empirically that on Stochastic Block Models along with mixture-of-Gaussian features, it is able to identify the clusters accurately. Further theoretical investigation into this standard setting and establishing error bounds for $S^3GC$ is of interest. $S^3GC$ can be applied to graphs with heterogeneous nodes, but it cannot explicitly exploit the information. Extension of $S^3GC$ to cluster graphs while directly exploiting heterogeneity of nodes is another open problem. Finally, $S^3GC$ like all deep learning methods is susceptible to being unfairly biased by a few "important" nodes. Ensuring stable clustering techniques with minimal bias for a small number of nodes is another interesting direction.

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
