# Appendix

This appendix is segmented into three key parts.

1. **Section A** discusses additional implementation details. In particular, an overview of the $S^3GC$ method is provided, method-specific overheads are discussed in detail and detailed hyper-parameter settings for our method and the main baselines reported in Table 3 are provided.

2. **Section B** reports additional dataset details, provides additional visualizations, ablation studies and main results with standard deviation values (not included in the main paper due to lack of space).

3. **Appendix C** provides theoretical analysis of the embedding produced by our method.

## A   Additional Implementation Details

### A.1   Overview of our method

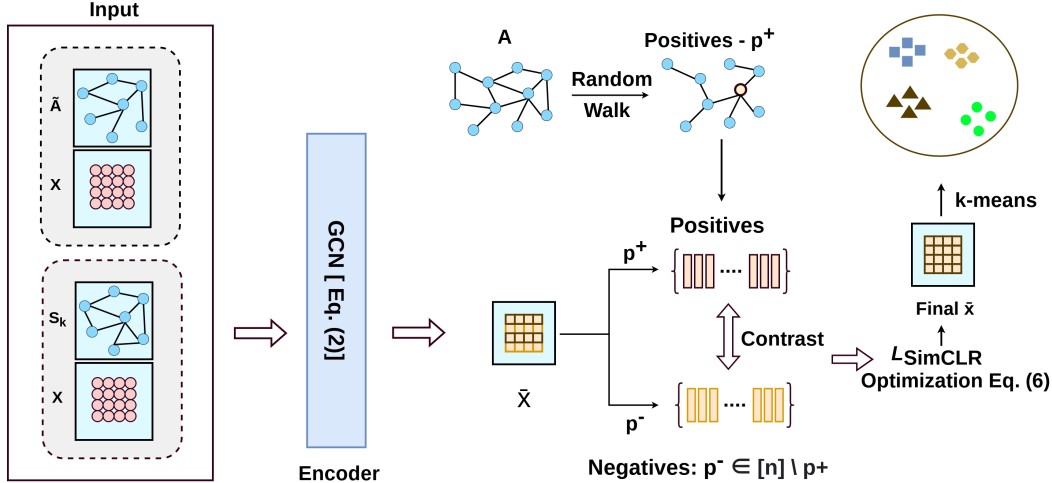

Figure 2: **Overview of the proposed method $S^3GC$**

We provide an overview of the architecture and training methodology of $S^3GC$ in Figure 2.

### A.2   Time and Memory Overheads for Various Methods

Table 6: **Time and Space Complexity of different methods**

| Method | Time Complexity (per epoch) | Space Complexity |
|---|---|---|
| MinCutPool | $\mathcal{O}(nk + m)$ | $\mathcal{O}(nk(n + k))$ |
| METIS | $\mathcal{O}(n + m + k log(k))$ | $\mathcal{O}(n + m)$ |
| Node2vec | $\mathcal{O}(rd)$ | $\mathcal{O}(nd)$ |
| DGI | $\mathcal{O}(md + nd^2)$ | $\mathcal{O}(m + nd + d^2)$ |
| DMoN | $\mathcal{O}(mk + nk)$ | $\mathcal{O}(m + nk)$ |
| GRACE | $\mathcal{O}(n^2d + md + d^2)$ | $\mathcal{O}(m + nd)$ |
| BGRL | $\mathcal{O}(md + nd^2)$ | $\mathcal{O}(m + nd + d^2)$ |
| MVGRL | $\mathcal{O}(n^2d + nd^2)$ | $\mathcal{O}(n^2 + nd + d^2)$ |
| **$S^3GC$** | $\mathcal{O}(nsd^2)$ | $\mathcal{O}(nd + rsd + d^2)$ |

In Table 6, we compare the time and space complexities of all the methods used in Table 3, and observe that $S^3GC$ performs better in terms of both time and memory complexity as compared to the other self-supervised learning methods that utilise both graph and feature information. Recall that $n$

is the number of nodes in the graph, $m$ is the number of edges, $d$ is the dimensionality of features, $r$ is the batch size, $k$ is the number of classes, and $s$ is the average degree per node.

For DGI, GRACE, and BGRL, the mentioned complexities are using full batch training. We use batched training on large datasets for DGI using GraphSAGE[19] which reduces time complexity to $O(nfd^2)$ and space complexity to $O(rfd + d^2)$, which is competitive with our method. Here, $f$ is the number of sampled neighbours in GraphSAGE per node.

### A.3 Hyperparameter Configurations for our method and the baselines

We use the k-means implementation from sklearn[4], METIS 5.1.0 from the official source[5] and Node2vec and DGI implementations from PyTorch geometric[6]. The sources for all the relevant baselines and their implementations are mentioned in Table 7

Table 7: **URL's and commit numbers to run baseline codes**

| Method | URL | Commit |
|---|---|---|
| MinCutPool | github.com/FilippoMB/MinCutPool | b27914 |
| Node2vec | github.com/pyg-team/pytorch_geometric/blob/master/examples/node2vec.py | 66b1780 |
| DGI | github.com/pyg-team/pytorch_geometric/blob/master/examples/infomax_inductive.py | 66b1780 |
| DMoN | github.com/google-research/google-research/tree/master/graph_embedding/dmon | 16b0d9c |
| GRACE | github.com/CRIPAC-DIG/GRACE | 51b4496 |
| BGRL | github.com/nerdslab/bgrl | dec99f8 |
| MVGRL | github.com/kavehhassani/mvgrl | 628ed2b |

We use the Adam optimizer for S$^3$GC and fix the embedding dimension to be the same across methods for a fair comparison, namely we set the embedding dimension $d$ to be 256 for all the methods for all datasets, except ogbn-papers100M where we use an embedding dimension $d = 64$ due to memory and scalability constraints. For all the methods we set the number of clusters equal to the number of classes. For trainable methods, a grid search was performed over hyperparameters specific to each method which is summarized below, while the other parameters are set to the default values:

1. **MinCutPool**: Learning Rate - {0.005, 0.001, 0.0005, 0.0001}, Num of Clusters = # of classes

2. **Node2vec**: Learning Rate - {0.01, 0.001}, Walk length - {10, 20, 40, 80 }, Context Size - {5, 10, 20, 40}

3. **DGI**: Learning Rate - {0.005, 0.001, 0.0005, 0.0001}, 3-hop Neighborhood sampling size (for large datasets) - {{15, 10, 5}, {25, 20, 10}}

4. **DMoN**: Learning Rate - {0.01, 0.005, 0.001, 0.0005, 0.0001}, Dropout - {0.0, 0.1, 0.2, 0.3, 0.4, 0.5}

5. **GRACE**: all hyperparameters as default provided by the authors for each dataset

6. **BGRL**: Learning Rate - {0.0005, 0.0001, 0.00005, 0.00001}, Dropout - {0.0, 0.1, 0.2, 0.3, 0.4, 0.5, 0.6}

7. **MVGRL**: Learning Rate - {0.005, 0.001, 0.0005, 0.0001}

8. **S$^3$GC**: Learning Rate - {0.01, 0.001}, Batch Size - {256, 512, 2048, 4096, 10000, 20000}, Walk Length - {3, 5, 10, 20, 50}, Number of walks per node - {10, 15, 20}

## B  Dataset Statistics and Additional Experimental Results

### B.1  Datasets

We use 7 datasets of three different scales of sizes, the statistics for which are provided in Table 2. We provide more information regarding the source and nature of the datasets as follows:

---

[4]https://scikit-learn.org/stable/modules/generated/sklearn.cluster.KMeans.html
[5]http://glaros.dtc.umn.edu/gkhome/metis/metis/download
[6]https://pytorch-geometric.readthedocs.io/

1. **Cora, Citeseer and Pubmed**[7]**:** These are three citation network datasets consisting of sparse bag-of-words feature vectors for each document. The edges denote citation links between the documents and are treated as undirected edges following the setup in [28]. Each node has a label class associated with the document.

2. **ogbn-arxiv**[8]**:** It is a citation dataset from the OGB node property prediction suite [25] representing the network of Computer Science ARXIV papers as indexed by MAG, where each node is a paper and each edge indicates a citation. Each node has an associated label which is one of the subject area from the 40 subject areas in ARXIV Computer Science papers. Feature vectors of the nodes are obtained from an average of word2vec [34] embeddings of the title and abstract.

3. **Reddit**[9]**:** The dataset is constructed from Reddit posts made in month of September 2014, with each node representing a post and node labels representing the community that the post belonged to. Nodes are connected based on common users commenting on both the posts, and node features are averaged 300-dimensional GloVe word vectors of the content associated with the posts such as title, comments, score and number of comments. More information regarding the setup can be found in [19].

4. **ogbn-products**[10]**:** It is an Amazon co-purchasing network dataset where nodes represent Amazon products and edges indicate that the two products are purchased together. Each node has an associated label which denotes the category of the product. Node features are dimensionality reduced bag of words features of the product descriptions, following the setup in OGB [25].

5. **ogbn-papers100M**[11]**:** It is a very large scale citation network dataset consisting of 111 million papers indexed by MAG. Node features and graph structure for this dataset is created in the same way as done for the ogbn-arxiv dataset in OGB [25], while labels are one of the 172 subject areas of a subset of papers published on ARXIV.

## B.2 Visualisation of embeddings for Synthetic Data Experiment

To understand the performance of $S^3GC$ as compared to the other methods in learning representations on Synthetic datasets, we observe the quality of the generated embeddings using t-SNE[51] projected in 2-dimensions for the setup discussed in Table 1 and Section 3.5. We note that the first row refers to a high quality graph with strong attributes, the second row refers to a weak graph with weak attributes, and the third row refers to a weak graph with strong attributes. We note that all methods perform similarly in the first setting when both the graph and attributes are of good quality, however show varied performance when either or both of them are varied in quality. Hence, Figure 3 visualizes the performance of Node2vec, DGI and $S^3GC$ in the second and the third settings. We observe from Fig. 3a that Node2vec doesn't learn much distinguishable representations when the quality of the graph is weak, which can be attributed to Node2vec being dependent only on the graph structure for learning embeddings. Note that we also perform an experiment with $S^3GC$ using only the weak graph information (without using any attributes), denoted by $S^3GC$-$\mathcal{I}$ and visualized in Figure 3b. We observe that $S^3GC$-$\mathcal{I}$ learns more "cluster-like" representations as compared to Node2vec even when utilizing only the weak quality graph information, indicating that the loss formulation in $S^3GC$ promotes learning clusterable representations. Then, we compare the performance of DGI and $S^3GC$ in the second and third settings, and visualize the learnt representations in Figures 3c to 3f. We observe that $S^3GC$ learns representations that correspond to more well-defined clusters as compared to DGI in each of the settings, indicating that $S^3GC$ is able to use both the graph and attribute information more efficiently, even in settings with varied data quality.

## B.3 Ablation on Batch Size

To understand the effect of varying the batch size in $S^3GC$, we perform an ablation study on the ogbn-arxiv dataset by keeping the other parameters such as learning rate and walk length constant,

---

[7]https://github.com/tkipf/gcn/tree/master/gcn/data
[8]https://ogb.stanford.edu/docs/nodeprop/#ogbn-arxiv
[9]http://snap.stanford.edu/graphsage/reddit.zip
[10]https://ogb.stanford.edu/docs/nodeprop/#ogbn-products
[11]https://ogb.stanford.edu/docs/nodeprop/#ogbn-papers100M

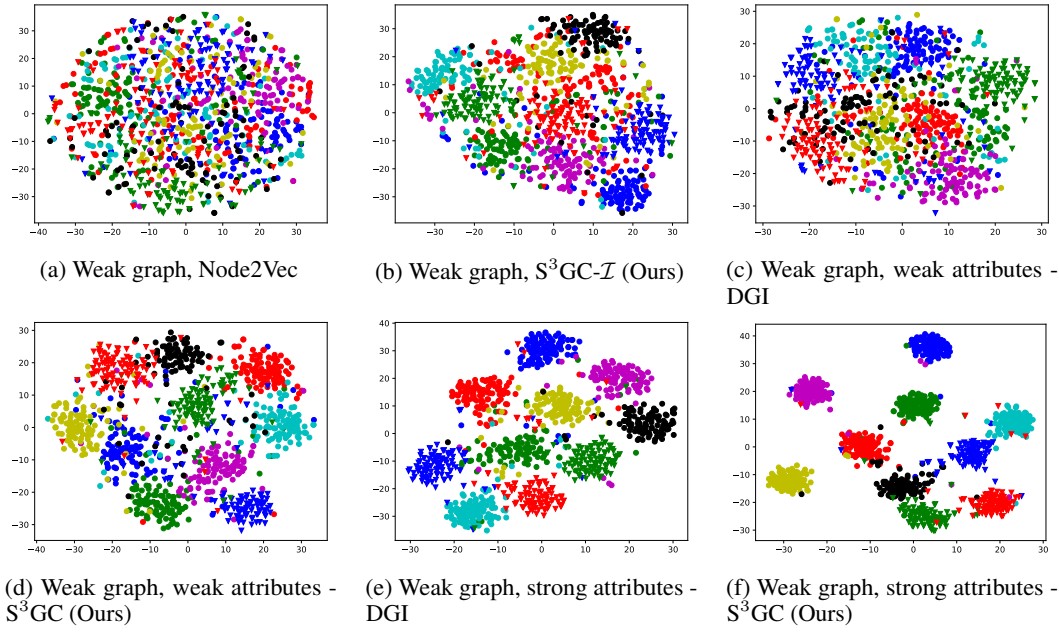

(a) Weak graph, Node2Vec

(b) Weak graph, S$^3$GC-$\mathcal{I}$ (Ours)

(c) Weak graph, weak attributes - DGI

(d) Weak graph, weak attributes - S$^3$GC (Ours)

(e) Weak graph, strong attributes - DGI

(f) Weak graph, strong attributes - S$^3$GC (Ours)

Figure 3: **Visualisation of embeddings.**

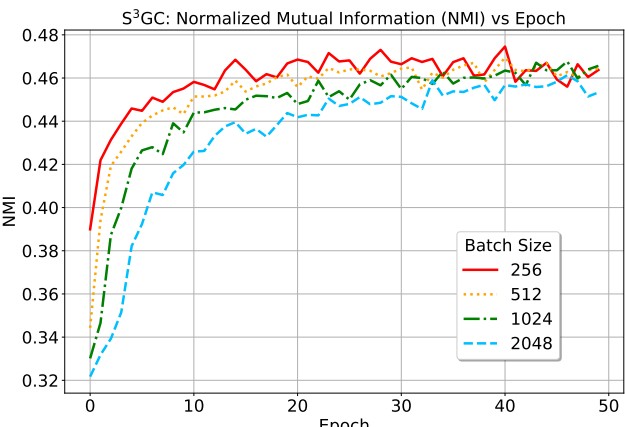

Figure 4: **Ablation study on the effect of using different batch sizes in S$^3$GC on the ogbn-arxiv dataset.**

while varying the batch size. We train S$^3$GC for different batch sizes from $\{256, 512, 1024, 2048\}$ and report the clustering NMI vs Epoch performance corresponding to each configuration in Figure 4. We observe that smaller batch sizes show faster convergence, and hence require lesser epochs to reach a reasonably good clustering performance in terms of NMI after which the performance saturates. Larger batches require more epochs however also require lesser per-epoch time as compared to smaller batches. We do note that the final performance corresponding to the different batch sizes are very similar.

### B.4  Ablation on Walk length

To understand the effect of varying the length of random walks in S$^3$GC for the sampling of positives, we perform an ablation study on the ogbn-arxiv dataset by keeping the other parameters such as learning rate and batch size constant, while varying the walk length. We train S$^3$GC for different walk lengths from $\{3, 5, 10, 20, 50, 100\}$ and report the clustering NMI vs Epoch performance corresponding to each configuration in Figure 5. We observe that smaller walk lengths upto $\sim 5$ show best performance in terms of the clustering NMI, after which the performance starts to degrade

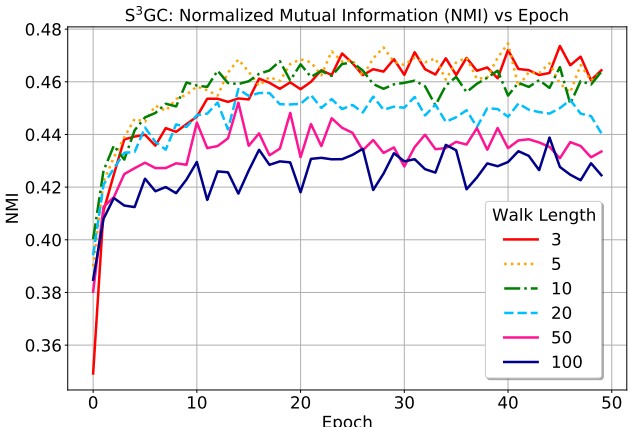

Figure 5: **Ablation study on the effect of using different walk lengths in S$^3$GC on the ogbn-arxiv dataset.**

with larger walk lengths. This can be attributed to the inclusion of unrelated or "farther-away" nodes belonging to different classes as positives in the batch. We also observe that the walk length parameter $\sim 5$ is optimal across datasets and hence does not require significant hyperparameter tuning.

### B.5 Main table results with mean and standard deviation values

We provide detailed results for all the methods with mean and standard deviation values in the evaluation of all the metrics across datasets, in Table 8 and Table 10 respectively. We observe similar results as discussed in the main paper.

Table 8: **Results of comparison of the embeddings generated by our method S$^3$GC as compared to different scalable methods on ogbn-papers100M with 111M nodes and 1.6B edges, with mean and std values**.

| Method | ogbn-papers100M | | | | |
| --- | --- | --- | --- | --- | --- |
| | Accuracy | NMI | CS | F1 | ARI |
| k-means | 0.144±0.004 | 0.368±0.004 | 0.342±0.003 | 0.101±0.004 | 0.074±0.007 |
| Node2vec | **0.175**±0.005 | 0.380±0.004 | 0.352±0.004 | **0.099**±0.009 | **0.112**±0.009 |
| DGI | 0.151±0.005 | 0.416±0.005 | 0.386±0.003 | **0.111**±0.010 | 0.096±0.008 |
| S$^3$GC (Ours) | **0.173**±0.004 | **0.453**±0.005 | **0.430**±0.003 | **0.118**±0.004 | **0.110**±0.007 |

### B.6 Ablation on the encoder components

We provide ablation on different components of the encoder used in S$^3$GC and their affect. The encoder has three parts: a) $\sigma(AX\theta)$, b) $\sigma(SX\theta')$, and c) $\mathcal{I}$, where $\sigma = PReLU$ activation. We study each combination's affect on Cora, Citeseet, Pubmed, ogbn-arxiv, Reddit, and ogbn-products in Table 9. We see that having only attribute information can cause suboptimal performance where attributes aren't strong enough, for example in Reddit dataset. While using only attribute information will not take benefit of attributes at all, like in Cora, Citeseer, ogbn-arxiv, and ogbn-products. The use of diffusion matrix $S$ seem to be more helpful in smaller datasets like Cora and Pubmed, but does provide marginal benefits in larger datasets like ogbn-arxiv and ogbn-products. In order to save memory, one can remove $\sigma(SX\theta')$ and still have reasonable good performance.

## C Theoretical analysis of embedding properties

A recent work [46] formalizes the concept of Positional and Structural embeddings, their properties, and their relation to each other. In this section, we follow the conceptual description from this work and prove that S$^3$GC produces positional embeddings rather than structural embeddings. The problem of community detection utilizes the close-knit behavior of nodes in one community, and

Table 9: **Results of different combinations of S³GC encoder**. The reported numbers are NMI on clustering $k$=#clusters of the respective dataset. $\sigma$ is PReLU activation

| Encoder | Cora | Citeseer | Pubmed | ogbn-arxiv | reddit | ogbn-products |
|---|---|---|---|---|---|---|
| $\mathcal{I}$ | 0.350 | 0.005 | 0.174 | 0.444 | 0.808 | 0.535 |
| $\sigma(AX\theta)$ | 0.561 | 0.430 | 0.331 | 0.457 | 0.775 | 0.533 |
| $\sigma(SX\theta')$ | 0.568 | 0.428 | 0.352 | 0.440 | 0.707 | 0.500 |
| $\sigma(AX\theta)+\sigma(SX\theta')$ | 0.572 | 0.423 | 0.330 | 0.460 | 0.777 | 0.528 |
| $\sigma(AX\theta)+\mathcal{I}$ | 0.575 | 0.426 | 0.327 | 0.459 | 0.810 | 0.531 |
| $\sigma(SX\theta)+\mathcal{I}$ | 0.580 | 0.431 | 0.330 | 0.460 | 0.806 | 0.540 |
| $\sigma(AX\theta)+\sigma(SX\theta)+\mathcal{I}$ | 0.588 | 0.441 | 0.333 | 0.463 | 0.807 | 0.536 |

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

hence positional embeddings encourage such nodes to be clustered together owing to their positional proximity in the original graph.

Before we prove our result, we introduce some basic notations, borrowing from [46]. Readers are encouraged to refer to [46] for a more detailed description of the definitions and the background.

We consider a graph $G = (V, X, E)$, where $V$ is the set of nodes and $|V| = n$, $E \in \mathbb{R}^{n \times n}$ is the set of edges in $V \times V$, and $X \in \mathbb{R}^{n \times d}$ are node attributes. Let $A \in \mathbb{R}^{n \times n}$ be the corresponding adjacency matrix.

**Definition 1** (Definition 2, [46] - Permutation Action $\pi$). *A permutation action $\pi$ is a function that acts on any vector, matrix, or tensor defined over the nodes $V$ , e.g., $(Z_i)_{i \in V}$ , and outputs an equivalent vector, matrix, or tensor with the order of the nodes permuted. We define $\Pi_n$ as the set of all $n!$ such permutation actions.*

**Definition 2** (Definition 3, [46] - Orbits). *An orbit is the result of a group action $\Pi_n$ acting on elements of a group corresponding to bijective transformations of the space that preserve some structure of the space. The orbit of an element is the set of equivalent elements under action $\Pi_n$, i.e., $\Pi_n(x) = \{\pi(x) | \pi \in \Pi_n\}$.*

**Definition 3** (Definition 4, [46] - G-equivariant and G-invariant functions). *Let $\Sigma_n$ be the set of all possible attributed graphs $G$ of size $n \geq 1$. More formally, $\Sigma_n$ is the set of all tuples $(A, X)$ with adjacency tensors $A$ and corresponding node attributes $X$ for $n$ nodes. A function $g : \Sigma_n \to \mathbb{R}^{n \times \cdot}$ is G-equivariant w.r.t. valid permutations of the nodes $V$ , whenever any permutation action $\pi \in \Pi_n$ in the $\Sigma_n$ space associates with the same permutation action of the nodes in the $\mathbb{R}^{n \times \cdot}$ space. A function $g : \Sigma_n \to \mathbb{R}^{n \times \cdot}$ is G-invariant whenever it is invariant to any permutation action $\pi \in \Pi_n$ in $\Sigma_n$.*

**Definition 4** (Definition 5, [46] - Graph orbits and Graph isomorphism)**.** *Let $G = (A, X)$ be a graph with $n$ nodes, and let $\Pi_n(G) = \{(A', X') : (A', X') = (\pi(A), \pi(X)), \forall \pi \in \Pi_n\}$ be the set of all equivalent (isomorphic) graphs under the permutation action $\pi$. Two graphs $G_1 = (A_1, X_1)$ and $G_2 = (A_2, X_2)$ are said isomorphic iff $\Pi_n(G_1) = \Pi_n(G_2)$.*

**Definition 5** (Definition 6, [46] - Node orbits & Node isomorphism)**.** *The equivalence classes of the vertices of a graph $G$ under the action of automorphisms are called vertex orbits. If two nodes are in the same node orbit, we say that they are isomorphic.*

Intuitively, Definition 5 says that two nodes are isomorphic if the "view" of the graph with respect to these nodes are the same. In other words, if the nodes' identities' are hidden, one can't distinguish between two isomorphic nodes as the graph structure would look exactly the same with respect to their positions.

**Definition 6** (Definition 8, [46] - Structural node representations)**.** *The structural representation of node $v \in V$ in a graph $G = (A, X)$ is the G-invariant representation $\Gamma(v, A, X)$, where $\Gamma : V \times \Sigma_n \to \mathbb{R}^d, d \geq 1$ such that $\forall u \in V, \Gamma((u, A, X)) = \Gamma(\pi(u), \pi(A), \pi(X))$ for all permutation actions $\forall \pi \in \Pi_n$. Moreover, for any two isomorphic nodes $u, v \in V, \Gamma(u, A, X) = \Gamma(v, A, X)$.*

Intuitively, Definition 6 implies that all isomorphic nodes (nodes which are structurally same) have the same embedding, and this embedding is independent of the permutation of nodes and node features input to the embedding function $\Gamma$. We now define positional embedding.

**Definition 7** (Definition 12 [46], (Positional) Node embedding)**.** *The (positional) node embeddings of a graph $G = (A, X)$ are defined as joint samples of random variables $(Z_i)_{i \in V}|A, X \sim p(\cdot|A, X), Z_i \in \mathbb{R}^d, d \geq 1$, where $p(\cdot|A, X)$ is a G-equivariant probability distribution on A and X, that is, $\pi(p(\cdot|A, X)) = p(\cdot|\pi(A), \pi(X))$ for any permutation $\pi \in \Pi_n$*

Definition 7 says that for a given node, the distribution of its embedding remains the same under permutation.

We now prove that S$^3$GC gives positional embeddings and not structural embeddings. For simplicity, let us asssume that given a fixed initialization of $\Theta, \Theta', \mathcal{I}$ (eq. (2)), the algorithm is deterministic. The assumption doesn't affect the final result, as the randomness over a random sampler would simply result in an additional expectation while giving the same final result.

First, notice that our model is permutation equivariant. Given $(A, X)$ or $(\pi(A), \pi(X))$ and a fixed initialization, our training will result in the embeddings being learnt as $\bar{X}$ and $\pi(\bar{X})$ respectively. This is because with permutations, the structure of the graph doesn't change. Hence, positive and negative node sampler will receive the same neighbourhood for any node, and would sample nodes in the same fashion. Also, the embedding output of encoder simply gets permuted:

$$\begin{aligned} Encoder =& PreLU(\pi(A)\pi(X)\Theta) + PreLU(\pi(S)\pi(X)\Theta') + \pi(\mathcal{I}) \\ =& \pi(PreLU(AX\Theta)) + \pi(PreLU(SX\Theta')) + \pi(\mathcal{I}) \\ =& \pi(PreLU(AX\Theta) + (PreLU(SX\Theta') + \mathcal{I}) \\ =& \pi(\bar{X}) \end{aligned}$$

Now, given the initialization $\Theta \sim \mathcal{N}(0, I), \Theta' \sim \mathcal{N}(0, I), \mathcal{I} \sim \mathcal{N}(0, I)$, a deterministic permutation equivariant function will result in permutation equivariant output distributions. Thus, we can say that $\pi(p(\cdot|(A, X))) = p(\cdot|\pi(A), \pi(X))$. Having randomness over a random sampler hence keeps the above property intact, owing to the permutation equivariant way of training.

We now show that our method doesn't produce structural embedding. We can take a toy example to demonstrate this. Let us consider a graph $G$ consisting of two disconnected cliques of equal size. Let's also say that $X = 0$. Therefore, our method will boil down to a contrastive learning framework similar to Node2Vec, where we initialize $\mathcal{I} \sim \mathcal{N}(0, I)$. But because any two nodes in the two different cliques are always negative samples of each other, the contrastive loss forces them to be dissimilar. Thus, structurally symmetric nodes in two different cliques do not learn the same embedding and hence our method does not produce structural embeddings.