# OpenReview forum: "S3GC: Scalable Self-Supervised Graph Clustering"
_NeurIPS.cc/2022/Conference — NeurIPS 2022 Accept_

### Official Review · Reviewer_8qP4 · 2022-07-08

**Rating:** 6
**Confidence:** 3
**Soundness:** 3 good
**Presentation:** 3 good
**Contribution:** 2 fair

**Summary:**

The paper presents a new method, S3GC, for scalable graph clustering. S3GC takes both the graph structure information and node-feature information into consideration by using a one-layer GNN encoder and performs self-supervised contrastive learning. There are several contributions. First, S3GC implements self-supervised learning on the graph clustering problem. Second, S3GC combines graph structure information and node feature information. Third, systematic experiments have been conducted.

S3GC has three key components, namely graph convolutional encoder, random walk sampler, and contrastive loss formulation. The graph convolutional encoder is a 1-layer Graph Convolutional Network, which is used to combine both graph structures and node features. One important design in the encoder is that the author uses mini-batches to reduce the memory requirement, which makes S3GC scalable to the large graph. Random walk sampler uses biased second order Random Walks with restarts to generate points similar, and thus capture the local neighborhood of each node. Contrastive loss formulation is a SimCLR style loss function, which is used for learning the encoder parameters. With this formulation, the authors use graph neighborhood information to produce augmentations of a node.

**Questions:**

Q1. Please check Line 214 and see whether "r" is missed.

Q2. Please report the details of running times for each method. As this paper studies scalability, these efficiency results are important.

Q3. In abstract, the authors state that "we demonstrate that S3GC is able to significantly outperform state-of-the-art methods in terms of clustering accuracy with as much as 5% gain in NMI". However, from Table 3 this seems not always true (e.g., compared with MVGRL). Please explain the reason.

Q4. The three components of S3GC use the existing designs partly. Please explain more exactly where the novelty lies.

**Limitations:**

This paper is in general elegant in terms of method design. However, the novelty is a bit unclear. All the three modules are inspired directly from the existing work. To make their combination a novelty, I believe a deeper discussion is needed, i.e., why such combinations can maintain the effectiveness. For example, 1-layer GCN seems to be simple, and why its combination with random walks and contrastive loss are effective?  Which part has the dominating factor? The related ablation study that checks the effectiveness of each component should be included in the main paper. The ablation study in the Appendix only includes some parameter testings.

**Strengths And Weaknesses:**


S1.	The paper is in general easy to follow.

S2.	Scalable clustering is important in practice.

S3.  The authors apply self-supervised learning to graph clustering, and present an effective and scalable model. The result shows that the model can address some existing limitations.

S4.	The proposed model can be applied to billion-edge graphs for clustering.

S5.	The author conducts extensive experiments and gives correlated analysis. The experimental result proves that S3GC can solve graph clustering problems with better performance in different aspects.

W1. The three components of S3GC all use the existing designs partly. The first part, the graph convolutional encoder, uses a 1-layer Graph Convolutional Network, and the novelty here seems to be the way of combining both graph information and node features. The second part, the random walk sampler, is inspired by previous work and formally follows previous work. The third part, contrastive loss formulation, uses a SimCLR style loss function.

W2. Seems there is a minor error in line 214. The author states that the time complexity is O(Knsd2), seemingly forgetting “r”.

W3. I did not find the results on detailed running times of different methods (though roughly mentioned).

---

> ### Author Response · Authors · 2022-08-02
> **Response to Reviewer 8qP4  - Thank you for the positive and thorough review. Part 1/2**
>
> We thank the reviewer for their comments. Please find attached our reply herewith.
>
> > **1. W1, Q4, Limitation
> W1: The three components of S3GC all use the existing designs partly. The first part, the graph convolutional encoder, uses a 1-layer Graph Convolutional Network, and the novelty here seems to be the way of combining both graph information and node features. The second part, the random walk sampler, is inspired by previous work and formally follows previous work. The third part, contrastive loss formulation, uses a SimCLR style loss function.
> Q4: The three components of S3GC use the existing designs partly. Please explain more exactly where the novelty lies.
> Limitation: This paper is in general elegant in terms of method design. However, the novelty is a bit unclear. All the three modules are inspired directly from the existing work. To make their combination a novelty, I believe a deeper discussion is needed, i.e., why such combinations can maintain the effectiveness. For example, 1-layer GCN seems to be simple, and why its combination with random walks and contrastive loss are effective? Which part has the dominating factor? The related ablation study that checks the effectiveness of each component should be included in the main paper. The ablation study in the Appendix only includes some parameter testings.**
>
>
> We would like to emphasize that the simplicity of our method is a strength and not a weakness as it makes our method more interpretable and the role of the individual design choices make it more intuitive. Though each individual component exists in literature, the careful design choices in using them together makes our method scalable and learns accurate clusterable representations. Our work is motivated by the theoretical properties of contrastive losses, however largely unexplored in the empirical setting of contrastive clustering and specifically graph clustering for which our work offers interesting insights. As you rightly pointed out, the uniqueness and novelty lies in the design choices put together. The strong empirical performance that we have demonstrated combined with the simplicity of the design choices would also make our method to be easily adoptable by practitioners and a wider community, as compared to the previous works.
>
> For example, the effectiveness of our loss function is evident from the experiments performed on the synthetic dataset based on SBM in Table-1. There, even when using only the graph structure information, we perform considerably better than Node2Vec. To show this effectiveness further, we perform experiments on the ogbn-arxiv dataset, and observe similar behavior. Node2Vec on ogbn-arxiv gets $0.41$ average NMI. While, $S3GC-\mathcal{I}$ (S3GC where the encoder is just a learnable embedding) gets $0.436$ average NMI.
> Similarly, using random walk to sample positive and negative pairs is important for scalability and to capture more than one-hop neighborhood. A random walk generator has been used in several embedding generation algorithms for the purpose of node information aggregation, as we have cited in our paper. Replacing this component with a different positive and negative sample generator but with the same encoder and loss formulation for S$^3$GC performs suboptimally: for instance, consider all the neighbors of a node as positive samples while negatives are chosen in a similar way sampled randomly. This yields suboptimal performance as we only take the immediate neighborhood as positive samples, whereas random-walk captures $k$-hop neighborhood. We provide numbers on ogbn-arxiv here for comparison. Our original proposed S$^3$GC formulation with the random walk based positive nodes sampler gets $0.463$ average NMI on the ogbn-arxiv dataset, whereas the latter proposed method in this paragraph of considering neighbors as positive samples gets $0.40$ average NMI.
>
> Finally, using 1-layer GCN encoder helps us in capturing local information scalably and accurately, whereas more complicated encoders can possibly help with improving performance further, we leave this as an avenue for future work. Replacing this component with just a learnable embedding tensor will not capture nodes attribute information, as we also saw in its performance on  the ogbn-arxiv dataset with a $0.436$ average NMI, as also reported in the previous paragraph.
>
> We agree that these are currently mentioned as finer points in the text and we will highlight an ablation with this information and more focused results on SBM and real graph datasets in the main paper to emphasize our novelty of the specific design choices we make, which results in our method being scalable, efficient, and clusterable.

---

> > ### Comment · Reviewer_8qP4 · 2022-08-07
> > **Response to authors**
> >
> > Thanks for the detailed feedback.
> >
> > I agree that simplicity is not weakness in many cases. My concern was not about the simplicity of the method, but lack of a strong experiment or theory to understand why such method can be effective.
> >
> > The response gives a better picture of the novelty of the paper and is more convincing. I hope these points about novelties can be incorporated in the future version of the paper.

---

> > > ### Author Response · Authors · 2022-08-08
> > > **Thanks for the review!**
> > >
> > > Thanks for the feedback. There exists theory studying the effectiveness of each of the S3GC components individually and in different settings [1,2,3,4], but none on studying them together in the context of graph clustering. In our work, as we discussed, we support our design choices and study their effectiveness empirically supporting them with thorough experiments. We hope to shed more theory on the combination of our design choices as a future work.
> > >
> > > Thanks for the discussion on bringing out the novelty aspect of our paper more clearly and convincingly and we will surely incorporate this in the final draft.
> > >
> > > [1] Semi-Supervised Classification with Graph Convolutional Networks
> > >
> > > [2] Convolutional neural networks on graphs with fast localized spectral filtering
> > >
> > > [3] node2vec: Scalable Feature Learning for Networks.
> > >
> > > [4] Provable Guarantees for Self-Supervised Deep Learning with Spectral Contrastive Loss

---

> ### Author Response · Authors · 2022-08-02
> **Response to Reviewer 8qP4 - Thank you for the positive and thorough review. Part 2/2**
>
> (Contd..)
>
> > **2. W2, Q1
> W2: Seems there is a minor error in line 214. The author states that the time complexity is O(Knsd2), seemingly forgetting “r”.
> Q1: Please check Line 214 and see whether "r" is missed.**
>
> The factor of $r$ gets cancelled. If there’s $n$ nodes, and each batch is of size $r$, then the total number of batches is $n/r$. Thus, if one batch takes $O(rsd^2)$ time, then each epoch will take time $O(n/r * rsd^2) = O(nsd^2)$. And $K$ epochs will take time $O(Knsd^2)$.
>
> > **3. W3, Q2
> W3: I did not find the results on detailed running times of different methods (though roughly mentioned).
> Q2: Please report the details of running times for each method. As this paper studies scalability, these efficiency results are important.**
>
>
> Thank you for the suggestion, we believe you mean the exact running times of each method. We have provided the theoretical time and space complexities for all the methods in Table 5 in the Appendix. Due to hardware and time constraints, we provide the actual running time of trainable methods on the OGBN-Products dataset, which is the second largest dataset that we have experimented with. We will provide the running times for all the methods across datasets in our final version.
>
> The below table provides the training times in seconds:
>
> | Dataset           | MinCutPool | DMoN   | DGI        | Node2Vec | S3GC |
> | ------------------  | --------------- |----------- | ----------- | -----------     | -------- |
> | ogbn-products | 3600           | 2653     |  1734     | 1819          | 3280   |
>
>
> > **4. Q3
> Q3: In abstract, the authors state that "we demonstrate that S3GC is able to significantly outperform state-of-the-art methods in terms of clustering accuracy with as much as 5% gain in NMI". However, from Table 3 this seems not always true (e.g., compared with MVGRL). Please explain the reason.**
>
> [Please also refer to our response to reviewer 4aEs on MVGRL’s performance.]
>
> MVGRL uses the full diffusion matrix in their methodology and experiments, while not using this diffusion matrix reduces the method to a DGI style base, which performs suboptimal as compared to S3GC as has been shown in our experiments. From this observation, we believe this could be one of the main reasons for MVGRL’s superior performance on smaller datasets where computing this style of diffusion matrix is possible. This is also the reason MVGRL does not scale to even moderately large graphs of the order of hundreds of thousands of nodes (because of using the diffusion matrix).
>
> ----
>
> We hope that the rebuttal clarifies questions raised by the reviewer. We would be very happy to discuss any further questions about the work, and would really appreciate an appropriate increase in score if reviewers’ concerns are adequately addressed to facilitate acceptance of the paper.

---

### Official Review · Reviewer_4aEs · 2022-07-11

**Rating:** 5
**Confidence:** 3
**Soundness:** 3 good
**Presentation:** 2 fair
**Contribution:** 2 fair

**Summary:**

This paper studies the problem of graph clustering using additional information such as node features. Previous works focus on learning graph embeddings and don’t explicitly optimize for clustering, are highly reliant on graph information which fall short when noisy or involve expensive modules which do not scale well. This method utilizes both node attribute information and the graph structure information to generate embeddings that are more linearly separable into k clusters. The experiments and results show that the proposed S3GC method outperforms several clustering baselines and produces better quality embedding clusters.

**Questions:**

1. In equation 5, both $\alpha$ and $alpha$ are used to indicate the same variable. It is important to be consistent with the notations.

2. In Table 1, the name S3GC-I is a little confusing because it usually implies S3GC method without (minus) I while it actually means using only I. So it would be good to consider changing the name to something more intuitive (eg. S3GC_I or Only I, etc)

3. The second observation under section “Setup and observations”, line 247, talks about DGI not performing well when attributes are noisy but does not reference any result table or figure. If the authors were referring to table 2, it is not clear how they have arrived at that conclusion. Which parameter controls the quality of the node attributes?

4. Similar comment for observation 3 (line 249),

5. Can the choice of parameter values (eg in table 1) be justified? The $p$, $q$ & $\sigma$ values chosen for the experiments are arbitrary and it is good to experiment with a bigger range of values. For eg, similar to Fig. 5a in [1].

6. Why does MVGRL perform better than S3GC in the small scale datasets? Is this just an artifact of the citation networks or is a similar behavior observed in other small scale datasets? It would be interesting to look at which components of MCGRL are contributing to its performance, especially on these datasets.

7. Gemsec [2] is another related work that has not been included in this paper but might be worth including in the baselines. It is similar in terms of learning clusterable graph embeddings. The embeddings are learned while considering the cluster information.

[1] - Grover, Aditya, and Jure Leskovec. "node2vec: Scalable feature learning for networks." Proceedings of the 22nd ACM SIGKDD international conference on Knowledge discovery and data mining. 2016.

[2] - Rozemberczki, Benedek, et al. "Gemsec: Graph embedding with self clustering." Proceedings of the 2019 IEEE/ACM international conference on advances in social networks analysis and mining. 2019.


**Limitations:**

The authors have mentioned some limitations of their work. It does not particularly consider different types of nodes in the learning of embeddings. It does not discount several “important” nodes that may cause the model to be unfairly biased.  However, there are several things that need to be improved in this paper (mentioned above) to make it a strong and novel contribution to the graph clustering literature.

**Strengths And Weaknesses:**

Strengths:
1. Related works are well covered.
2. Proposed method is simple and neat. It scales well to large graphs of the order of 100 million nodes.
3. The standard experiment setup has been followed. Synthetic data generation uses the well-studied SBM algorithm.
4. Experiments on large scale datasets show the superior performance of the proposed method in comparison with the baselines.
5. The paper is written in a simple and easy-to-follow manner.

Weaknesses:
1. The novelty in the proposed method is weak. It is rather similar to Node2Vec but with the SimCLR loss function which improves linear separability in the node representations.
2. Considering that there are several hyperparameters involved in the proposed algorithm, the paper does not include a systematic study of different ranges for the parameter values.

---

> ### Author Response · Authors · 2022-08-02
> **Response to Reviewer 4aEs - Thank you for your review -  Part 1/2**
>
> We thank the reviewer for their time and comments. Please find our reply herewith.
>
> > **1. The novelty in the proposed method is weak. It is rather similar to Node2Vec but with the SimCLR loss function which improves linear separability in the node representations.**
>
> We humbly disagree with the comment and would like to counter with several arguments.
>
> Firstly, the only common element between S3GC and Node2Vec is the random walk generator which has been used in several other embedding generation algorithms for the purpose of node information aggregation, as we have cited in our paper. In contrast to this, we only use the random walk information for creating the positives whereas our encoder and the loss formulation are different. Hence the statement that S$^3$GC is similar to only Node2vec with a contrastive loss formulation is an overly simplified generalization.
>
> Secondly, we argue that the novelty is manifold and more subtle than the above simple description of our method. We believe that the simplicity of our method is a strength and not a weakness as it makes our method more interpretable and the role of the individual design choices make it more intuitive. The specific design choices for our method S$^3$GC are carefully chosen in a way that they cater to the clustering objective to learn accurately clusterable representation while being simultaneously scalable. The strong empirical performance that we have demonstrated combined with the simplicity of the design choices would also make our method to be easily adoptable by practitioners and a wider community. Please also refer to our reply to Reviewer 8qP4 under “W1, Q4, Limitation” section for more discussion on the same.
>
> Finally, a lot of methods may seem “similar” to existing methods but involve detailed and important differences. For example, MVGRL (https://github.com/kavehhassani/mvgrl) can practically be regarded as sum of two DGI(s) (https://github.com/PetarV-/DGI), one with DGI(A, X) and another with DGI(S, X), where S is the diffusion matrix. However we see it as a valuable contribution, based on the practical insights that it brought in the community. We would like to argue that our paper would be a suitable contribution in the space of both large scale clustering and contrastive learning and would be useful not only for the graph community to motivate more works in this practical domain of graph clustering, but the insights will also inspire general work in both theoretically and empirically motivated ideas at the intersection of contrastive learning and clustering.
>
> > **2. Considering that there are several hyperparameters involved in the proposed algorithm, the paper does not include a systematic study of different ranges for the parameter values.**
>
> As mentioned in the paper in lines 300-312 and lines 343-351, there are actually very few hyperparameters used in S$^3$GC. Walk length, batch size, and number of epochs are the only three and important hyperparameters that we use. The rest (p, q in random walk) and $\alpha_i$ values for diffusion matrix are chosen as the standard values, $p=q=1$, that have been used in literature resulting in a non-biased random walk, and $\alpha_i$ as per the PPR equation (Page et al., 1999). We discuss the used hyperparameters in L343 with more details in the appendix and we do not foresee any significant changes to the performance of our method on varying these values. However for completeness, we will add a study on the varying the random walk parameters as well to the final version.
>
> > **Questions
> 1: In equation 5, both \alpha and alpha are used to indicate the same variable. It is important to be consistent with the notations.
> 2: In Table 1, the name S3GC-I is a little confusing because it usually implies S3GC method without (minus) I while it actually means using only I. So it would be good to consider changing the name to something more intuitive (eg. S3GC_I or Only I, etc)**
>
> Thank you for pointing this out, we will correct these typos as suggested.

---

> > ### Comment · Area_Chair_A6b3 · 2022-08-06
> > **AC comment**
> >
> > Authors: "whereas our encoder and the loss formulation are different" => how are they different? It is important to be specific when addressing a reviewer's concerns. How different? Where is the difference in the paper?

---

> > > ### Author Response · Authors · 2022-08-08
> > > **Reply to the AC comment**
> > >
> > > Thanks for bringing this to our notice, we will be more specific in our replies.
> > >
> > > S3GC encoder and loss formulation are different compared to Node2Vec. The encoder in Node2Vec is just a learnable embedding, thus not capturing any attribute information. While the S3GC encoder is a combination of single-layer GCN using adjacency matrix, capturing local attribute information; a single-layer GCN using diffusion matrix, capturing more global-view information; and a learnable embedding capturing graph structure information.
> > >
> > > For the loss function, Node2Vec uses NCELoss, whereas we use the SimCLR loss. We also show a difference in performance of these loss functions on synthetic SBM datasets.

---

> > ### Comment · Reviewer_4aEs · 2022-08-06
> > **Updating review scores**
> >
> > I thank the authors for their detailed responses and clarifications.
> >
> > I agree that the simplicity of your method is a strength and not a weakness and your response to W1 is much more convincing than what is discussed in the paper. As reviewer 8qP4 pointed out, the novelty lies in the design choice of putting together different components and I think it is important that this be highlighted in the paper. I would suggest including the same arguments to bolster your claims of novelty in the paper.
> >
> > Thank you for considering the suggestion of including Gemsec in your experiments. Since their paper seems to show improved performance over Node2Vec, it would be interesting to see their performance in comparison to S3GC.
> >
> > Almost all other concerns of mine have been addressed and I would like to update my review score.

---

> > > ### Author Response · Authors · 2022-08-08
> > > **Thanks for the updated review!**
> > >
> > > Thanks for your comments. We will surely highlight the novelty aspect of our method much more clearly.
> > >
> > > Regarding performance comparison of S3GC with Gemsec: We ran gemsec on Cora and Citeseer and provide a summary of the results and comparison with S3GC and other methods in our comment in response to the AC above. We observe that Gemsec performs comparably with other graph-only based methods, such as Node2vec, however performs suboptimally as compared to S3GC which uses both graph and feature information.
> > > We would be happy to engage in further discussions. We hope we now clarified all the concerns, and if so, would request your support in favor of our work by an appropriate rating for our work. Thanks again!

---

> ### Author Response · Authors · 2022-08-02
> **Response to Reviewer 4aEs - Thank you for your review - Part 2/2**
>
> (Contd..)
>
> > **3. The second observation under section “Setup and observations”, line 247, talks about DGI not performing well when attributes are noisy but does not reference any result table or figure. If the authors were referring to table 2, it is not clear how they have arrived at that conclusion. Which parameter controls the quality of the node attributes?
> 4. Similar comment for observation 3 (line 249),**
>
> The paragraph “Setup and Observations” starting L238 discusses Table 1, and what parameters control the quality of attributes and graph structure. As mentioned in L240, ratio $\sigma_c^2/\sigma^2$ controls the quality of node attributes (as also mentioned in L226). And hence, as mentioned in L241, the second row in Table 1 represents an instance having low node attributes, on which DGI performs sub-optimally. One can see the attributes have less quality by the performance of standard k-means on this instance. We will make this description more verbose for clarity.
>
> > **5. Can the choice of parameter values (eg in table 1) be justified? The p, q & \sigma values chosen for the experiments are arbitrary and it is good to experiment with a bigger range of values. For eg, similar to Fig. 5a in [1].**
>
> We choose $p, q, \sigma$ values so as to create instances having a mix of comparatively/relatively strong/weak graph structure and strong/weak node attributes. The metric to check graph structure quality is performance of classic Spectral Clustering (referred to as SC in Table-1). And the metric to measure node attribute quality is performance of standard k-means on the attributes, also measured in Table-1.
> While one can vary $p,q,\sigma$ to create instances having more weak/strong graph structure and node attributes, our focus has just been to show relative performances and we expect the trend to be the same with different absolute values.
>
> > **6. Why does MVGRL perform better than S3GC in the small scale datasets? Is this just an artifact of the citation networks or is a similar behavior observed in other small scale datasets? It would be interesting to look at which components of MCGRL are contributing to its performance, especially on these datasets.**
>
> MVGRL performs better because in addition to using an adjacency matrix, which provides a local view of the graph, it also uses a diffusion matrix providing a more global view of the graph. So while this helps with performance for the smaller datasets, it also makes the MVGRL method non-scalable coming from computing the entire diffusion matrix. MVGRL aims to build a different view of the graph structure through the diffusion matrix, hence needs an entire $n \times n$ matrix. Note that it can’t precompute some approximate version of $SX$ where $S$ is the diffusion matrix as well. This is because the architecture requires computing $SX'$ in every train step, where $X'$ is a noisy version of the attribute matrix $X$. Hence, the $S$ matrix (or approximate version of it) is required to be available in every epoch, making the non-scalability of MVGRL an issue in their methodology, while our aim is to specifically achieve both scalability and high clustering accuracy which is often the intended outcome in the real world setting for large graphs. As the reviewer pointed out, it would be an interesting ablation on MVGRL to confirm our hypothesis for their better performance for smaller datasets, however owing to the short rebuttal timeline, we defer this to future work.
>
> > **7. Gemsec [2] is another related work that has not been included in this paper but might be worth including in the baselines. It is similar in terms of learning clusterable graph embeddings. The embeddings are learned while considering the cluster information.**
>
> Thank you for pointing out this interesting work, we will include this in Related Works under the section describing graph only methods and are working on experimenting with the open sourced implementation to add empirical comparisons. Since the code is from 2018 and uses significantly outdated dependencies as well as the datasets used are all different from the ones we experiment with, we are working on getting these numbers as soon as possible and will update them soon. Since the method utilizes only the graph information and ignores the node attributes altogether, we expect the performance of the method to be very similar to Node2vec, facing significant degradation in the clustering quality when the graphs are noisy but node features are highly informative.
>
> ----
>
> We hope that the rebuttal clarifies questions raised by the reviewer. We would be very happy to discuss any further questions about the work, and would really appreciate an appropriate increase in score if reviewers’ concerns are adequately addressed to facilitate acceptance of the paper.

---

> > ### Comment · Area_Chair_A6b3 · 2022-08-06
> > **AC comment to authors and reviewer**
> >
> > Authors: What is the status of the Gemsec results?
> >
> > Reviewer 4aEs: The authors' answers seem detailed. Does it address most of your concerns?

---

> > > ### Comment · Reviewer_4aEs · 2022-08-06
> > > **Authors' answers address most concerns**
> > >
> > > Yes, most of my concerns have been addressed and I have updated my score.

---

> > > ### Author Response · Authors · 2022-08-08
> > > **GEMSEC performance**
> > >
> > > We were able to get Gemsec running. As expected, since gemsec uses only graph structure information, it performs suboptimal to S3GC but comparable to graph-structure only methods. We provide nmi numbers here on the Cora and Citeseer datasets, and compare with other graph-structure only methods in addition to S3GC.
> > >
> > > | Dataset             |  METIS  | Node2Vec  | GEMSEC  | S3GC  |
> > > | -------------------- | ----------- |-----------       |  -----------   |---------- |
> > > | Cora                 |  .396      | .444            |  .446          | .588     |
> > > | Citeseer           |  .170      |  .240            |  .185         | .441     |

---

### Official Review · Reviewer_2dNT · 2022-07-15

**Rating:** 6
**Confidence:** 4
**Soundness:** 3 good
**Presentation:** 3 good
**Contribution:** 2 fair

**Summary:**

The paper proposes a Scalable Self-Supervised method that uses graph neural networks (hence called S3GC) and node features to learn clusterable representations using contrastive learning.

S3GC uses a 1 layered graph convolutional network to encode feature and structural information. To encode the structure, the authors propose to use normalized adjacency and a k-hop diffusion matrix. To use the contrastive loss for generating clusterable representations, positive and negative samples are needed. S3GC uses a biased random walk sampler to get similar nodes. Negative nodes are generated randomly from the remaining nodes.

Authors have shown that it can scale well to graphs with 100M nodes which are challenging for current graph neural network-based approaches. S3GC is shown to gain as much as 5% in NMI.

**Questions:**

Is there an ablation that uses only the normalized adjacency/diffusion matrix in the GCN.  I didn’t find these in the paper (or in the appendix).

This paper refers to DGI as SOTA (page 6 lines 233-235, 273-274). Is that true? Perhaps the authors meant to say DGI style methods? MVGRL has been shown to work better and that is validated in the experiments as well (Table 3). Thus, I think Table 1 should have results with MVGRL. I understand authors have concerns that the reported values are higher in MVGRL (line 317-318) but I think it is good to add for completeness.

**Limitations:**

See Weaknesses +  questions.

**Strengths And Weaknesses:**

**Strengths**

The proposed approach is simple and intuitive.

The paper is well written and easy to understand.

The proposed approach combines information from both the graph and the node features and hence is applicable in settings where one vs the other is noisy. Extensive experiments are conducted to validate this using a synthetic dataset.

This method only uses a single GCN layer with normalized adjacency and diffusion matrix and thus it can take into account higher order neighborhood information and still scale to very large-scale graphs which allows for its application in real-world settings.

Extensive experimentation is done to show the effectiveness of the approach on small-scale as well as large-scale datasets.
Space and time complexity provided add to the clarity of the central idea of the paper about scalability.

**Weaknesses**

IMO this is more of applied work and I think perhaps an applied conference is a better place for this work for a larger impact. All the insights for instance for linear separability and clusterable contrastive representations are known ([1]). The part about scaling that uses a diffusion matrix has also been used as a scalable way of getting higher-order information ([2]). I think perhaps (to the best of my knowledge) the idea of enforcing closeness of neighbors is new but it doesn’t contribute to the main idea of the paper which is scalability.

It will be interesting to see a baseline that compares the baseline contrastive learning approaches (DGI, MVGRL, etc.) with just one gcn layer with 1 hop neighborhood perturbations. I think MVGRL will become pretty scalable in that setting as well and because it also uses a diffusion matrix my hunch is that it will be pretty competitive on larger datasets. From table 3 it seems like MVGRL is better than the proposed approach on average (on NMI at least which is considered to be the metric for clustering) whenever it didn’t have OOM error.

Another potential weakness is that all of the datasets are homophilous. I am not sure how will this perform on heterophilous graphs.

Another weakness is that it is not clear how this approach is directly applicable to heterogeneous graphs (which authors mention as well).

As authors point out, there is no theoretical justification for the proposed method. Perhaps one can borrow ideas from [1]

[1] Provable Guarantees for Self-Supervised Deep Learning with Spectral Contrastive Loss

[2] SIGN: Scalable Inception Graph Neural Networks

---

> ### Author Response · Authors · 2022-08-02
> **Response to Reviewer 2dNT - Thank you for the positive and thorough review. Part 1/2**
>
> We thank the reviewer for their helpful comments. Please find our reply herewith.
>
> > **1. IMO this is more of applied work and I think perhaps an applied conference is a better place for this work for a larger impact**
>
> - We humbly disagree with the reviewers’ comment here, NeurIPS has had a long track record of publishing and highlighting significant applied research contributions with a strong empirical impact for the broader ML community; we argue that our paper would be a suitable contribution in the space of both large scale clustering and contrastive learning as well and would be useful not only for the graph community to motivate more works in this practical domain of graph clustering, but the insights will also inspire general work in both theoretically and empirically motivated ideas at the intersection of contrastive learning and clustering.
>
> > **2. All the insights for instance for linear separability and clusterable contrastive representations are known ([1])**
>
> - We agree that theoretical insights showing effectiveness of contrastive loss on clustering has been shown in [1]. However we would like to argue that our work highlights its practical implications, application and evaluation in a real-world setting. It is often the case that theoretical results do not generalize well when experimented with in real world (especially large scale) datasets. Even in [1], authors show theoretical guarantees on a more general contrastive loss formulation, while our work analyses its effectiveness in practice where we observe that some contrastive formulations perform better than the others depending on the task. For example, consider the clustering experiments on synthetic datasets based on SBM, In Table 1, we observe that when only using the graph structural information, SimCLR based contrastive loss performs considerably better than Node2Vec loss, even though both of them can be considered variants of contrastive loss formulations. These insights can be very useful in designing more efficient and accurate clustering methods that work on large datasets in practice.
>
> > **3. The part about scaling that uses a diffusion matrix has also been used as a scalable way of getting higher-order information ([2]).**
>
> - While we agree that ideas like diffusion matrices have been used for computing higher order information before even before SIGN, such as PPRGo [3] and APPNP [4] for example, we would like to argue that the specific design choices that allows our method S$^3$GC to be highly scalable are not limited to only the diffusion matrix, but a careful combination of several specific ideas which makes our algorithm scalable and unique. Please also refer to our reply to Reviewer 8qP4 for more information in this regard. Thanks!
>
> [3] Scaling Graph Neural Networks with Approximate PageRank, Bojchevski et. al, SIGKDD 2020
>
> [4] Predict then Propagate: Graph Neural Networks Meet Personalized PageRank, Gasteiger et. al, ICLR 2019
>
> > **4. I think perhaps (to the best of my knowledge) the idea of enforcing closeness of neighbors is new but it doesn’t contribute to the main idea of the paper which is scalability.**
>
> - The idea of enforcing closeness of neighbors doesn’t contribute to scalability, but it is important for learning accurate and clusterable representations, hence important to the task of clustering. The central theme of the paper is scalably performing “accurate” clustering, which is what differentiates our work from something scalable like Node2Vec, which agreeably scales to large datasets, but does not generate representations that are well clusterable, which is evident from the main empirical results in our paper. Hence, we would like to argue that the general theme of the paper is both scalability and learning accurate representations for clustering, and enforcing closeness contributes significantly to the latter.

---

> > ### Comment · Reviewer_2dNT · 2022-08-05
> > **Follow-ups**
> >
> > I thank the authors for answering my questions and clarifications. I have some follow-up questions:
> >
> > > Thank you for the suggestion. Unless the dataset is relatively small like Cora, Citeseer, Pubmed, all of the existing methods perform better as the depth of the GCN increases. Therefore, the numbers reported for the baselines won’t get better when GCN depth is reduced to 1. For MVGRL, the non-scalability comes from computing the entire diffusion matrix. MVGRL aims to build a different view of the graph structure through the diffusion matrix, hence needs an entire  matrix. Note that it can’t precompute some approximate version of  where  is the diffusion matrix as well. This is because the architecture requires computing  in every train step, where  is a noisy version of the attribute matrix . Hence, the  matrix (or approximate version of it) is required to be available in every epoch, making the non-scalability of MVGRL an issue in their methodology altogether and something that we don’t foresee getting resolved with a more efficient implementation since it will always run into OOM errors as compared to all the other methods.
> >
> > Computing a diffusion matrix is 1-time operation. Once computed can be used as an adjacency matrix. There has been some work that precalculates the diffusion matrix even for graphs of the size of ogbn-products [1]. The diffusion matrix as the authors allude to this as well can be sparsified if it is dense. Since this is a one-time operation one can do this on the CPU. Can the authors help me understand why this is not scalable/possible? Also, IIUC MVGRL perturbs the diffusion matrix as well and not just the node features.
> >
> > > We humbly disagree with the reviewers’ comment here, NeurIPS has had a long track record of publishing and highlighting significant applied research contributions with a strong empirical impact for the broader ML community; we argue that our paper would be a suitable contribution in the space of both large scale clustering and contrastive learning as well and would be useful not only for the graph community to motivate more works in this practical domain of graph clustering, but the insights will also inspire general work in both theoretically and empirically motivated ideas at the intersection of contrastive learning and clustering.
> >
> > I would really appreciate if the authors substantiate this with evidence for a graph learning community where the insights were already known and the publication showed an application to large-scale settings. IMO this work is significant applied work for large-scale settings but as it stands I see this paper as a combination of several interesting insights that were known before which authors have also agreed to (at least I hope they do).
> >
> >
> > [1] SIGN: Scalable Inception Graph Neural Networks

---

> > > ### Author Response · Authors · 2022-08-08
> > > **Reply to the follow-ups (1/2)**
> > >
> > > > **Computing a diffusion matrix is 1-time operation. Once computed can be used as an adjacency matrix. There has been some work that precalculates the diffusion matrix even for graphs of the size of ogbn-products [1]. The diffusion matrix as the authors allude to this as well can be sparsified if it is dense. Since this is a one-time operation one can do this on the CPU. Can the authors help me understand why this is not scalable/possible? Also, IIUC MVGRL perturbs the diffusion matrix as well and not just the node features.**
> > >
> > > Thanks for the clarification, we understand your question better now. MVGRL originally uses a non-sparse full $n \times n$ sized diffusion matrix S, hence as such it is non-scalable. We also mentioned that an approximate version of S cannot be precomputed where we meant to refer to an approximation of $S=\sum_{i=1}^{\infty} \tilde{A}^i$ by $S’=\sum_{i=1}^{k} \tilde{A}^i$ for some finite k, as we use in our work (calling it k-hop diffusion matrix). This approximation is not possible since S’ would be dense too (even for finite small and finite k), and MVGRL needs $S’$ in every training epoch to multiply with the noised attribute matrix X'.
> > >
> > > We agree though, that a sparse-approximation is possible. Previous works like [1] also pre-compute this sparse diffusion matrix, and not the full dense matrix. Following [1,2] as per your suggestion and the set of experiments you recommended in your original comment, we ran the following experiments using sparse approximation of diffusion matrix for MVGRL, to scale to larger graphs and provide performance comparisons on the Cora and ogbn-arxiv datasets. We chose the Cora dataset to gauge the effect of the approximation on the quality of the embeddings and the ogbn-arxiv to benchmark against our method on a moderate-sized graph, where MVGRL wasn’t originally scalable.
> > >
> > > For Cora, we get a similar close nmi after the sparse approximation compared to using full-scale diffusion matrix as in original mvgrl
> > > For arxiv using 1-layer GCN to capture immediate neighbourhood, sparse-mvgrl gives a performance boost over DGI, but still performs suboptimal to S3GC. Using 2-layer gcn though didn’t improve the performance of mvgrl further over 2-layer DGI, while does improve the performance of 2-layer DGI over its 1-layer variant. For Cora, best performance is only using a single layer, hence we report that. The following reports mean nmi numbers of the experiments:
> > >
> > > | Dataset                           | DGI             | sparse-mvgrl   | S3GC   |
> > > | --------------------------        | ---------------  |-----------            | -----------|
> > > | Cora                                | .5710          | .6010               | .5880    |
> > > | ogbn-arxiv - 1 layer gcn  | .3701          | .4013               | .4663    |
> > > | ogbn-arxiv - 2 layer gcn  | .4180          | .4120               |  N/A    |
> > >
> > > The full diffusion matrix is of size $n\times n$. For the arxiv dataset the sparsification of the diffusion matrix results in 9837767 non-zero entries, equivalent to having roughly 4918883 edges (which is around 3.6 times the original number of edges in the original graph). Thus, the quality of the approximation should be good, as it should capture much more than just the immediate neighbourhood.
> > >
> > >
> > > For reproducibility, we provide the code for sparse-mvgrl in the following google drive link (anonymized) under “sparse mvgrl” folder: https://drive.google.com/drive/folders/18B_eWbdVhOURZhqwoBSsyryb4WsiYLQK?usp=sharing
> > >
> > > We run the sparse-mvgrl benchmark using different hyperparameters and sparsification methods and provide files with the best configuration. The DGI numbers reported in the above table can also be reproduced using the same code by uncommenting the 2 commented out lines. In case the program doesn’t fit in the memory, please use a higher ‘eps’ for the arxiv files and smaller ‘k’ values for Cora in the sparsification step.
> > >
> > > ------
> > > We hope this addresses your concern. We'll be happy to answer any further follow-ups.
> > >
> > > [1] SIGN: Scalable Inception Graph Neural Networks
> > >
> > > [2] Diffusion Improves Graph Learning

---

> > > > ### Comment · Reviewer_2dNT · 2022-08-09
> > > > **Updating my score.**
> > > >
> > > > I thank the authors for reasonably responding to my queries and providing a detailed response. I have updated my score accordingly.

---

> > > ### Author Response · Authors · 2022-08-08
> > > **Reply to the follow-ups (2/2)**
> > >
> > > Contd..
> > >
> > > > **I would really appreciate if the authors substantiate this with evidence for a graph learning community where the insights were already known and the publication showed an application to large-scale settings. IMO this work is significant applied work for large-scale settings but as it stands I see this paper as a combination of several interesting insights that were known before which authors have also agreed to (at least I hope they do).**
> > >
> > > Thank you for your suggestion. While there are several works in the graph learning community which build on top of insights that are previously known and help in developing either generalized framework or scale to larger datasets, there is one such work that we find particularly interesting and would like to mention: GNNAutoScale [1] by Fey et. al, published at ICML 2021. The primary contribution of the work is to propose a generalized framework for scaling existing message passing based GNN operators such as GCN, GAT, APPNP, GCNII, etc to very large graphs, the foundation of which is the methodology of approximation using historical embedding proposed in VR-GCN [2]. GNNAutoScale achieves scalability by pruning the computation graph of historical embeddings, relying on METIS for clustering, and implements efficient GPU–memory interfacing for scalable access.
> > >
> > > We believe that even though the idea and insights for different components were known from before: using existing GNN operators such as GCN and GAT, historical activations specifically in the context of graph convolutions were discussed in multiple works preceding this paper (VR-GCN, MVS-GNN) and METIS based clustering was explored in ClusterGCN [3], the proposed methodology using a combination of these ideas offered scalability which benefits the graph learning community in tackling similar challenges.
> > >
> > > In summary, there do exist interesting insights from existing works. But to choose the right components in the design choice, so that it works both efficiently and scalably well is a novel contribution overcoming many challenges which don’t seem apparent at first sight. Moreover, there are no previous works scaling graph clustering up to 100M nodes. Hence, our study incorporates studying existing methods, finding the challenges, and overcoming them with correct design choices. As we also mention in our response to Reviewer 8qP4, many other intuitive design choices don’t work as well.
> > >
> > > Based on these grounds, we feel our work makes many novel contributions in terms of both scaling (as explained in our responses) as well as to clustering (as you rightly pointed out). We sincerely hope this provides a satisfactory justification to the reviewer.
> > >
> > > [1] GNNAutoScale: Scalable and Expressive Graph Neural Networks via Historical Embeddings, Fey et. al, ICML ‘21
> > >
> > > [2] Stochastic Training of Graph Convolutional Networks with Variance Reduction, Chen et. al, ICML ‘18
> > >
> > > [3] ClusterGCN: An Efficient Training Algorithm for Training Deep and Large Graph Convolutional Networks, Chiang et. al, KDD ‘19

---

> ### Author Response · Authors · 2022-08-02
> **Response to Reviewer 2dNT - Thank you for the positive and thorough review. Part 2/2**
>
> (Contd..)
>
> > **5. It will be interesting to see a baseline that compares the baseline contrastive learning approaches (DGI, MVGRL, etc.) with just one gcn layer with 1 hop neighborhood perturbations. I think MVGRL will become pretty scalable in that setting as well and because it also uses a diffusion matrix my hunch is that it will be pretty competitive on larger datasets. From table 3 it seems like MVGRL is better than the proposed approach on average (on NMI at least which is considered to be the metric for clustering) whenever it didn’t have OOM error.**
>
> - Thank you for the suggestion. Unless the dataset is relatively small like Cora, Citeseer, Pubmed, all of the existing methods perform better as the depth of the GCN increases. Therefore, the numbers reported for the baselines won’t get better when GCN depth is reduced to 1. For MVGRL, the non-scalability comes from computing the entire diffusion matrix. MVGRL aims to build a different view of the graph structure through the diffusion matrix, hence needs an entire $n \times n$ matrix. Note that it can’t precompute some approximate version of $SX$ where $S$ is the diffusion matrix as well. This is because the architecture requires computing $SX'$ in every train step, where $X'$ is a noisy version of the attribute matrix $X$. Hence, the $S$ matrix (or approximate version of it) is required to be available in every epoch, making the non-scalability of MVGRL an issue in their methodology altogether and something that we don’t foresee getting resolved with a more efficient implementation since it will always run into OOM errors as compared to all the other methods.
>
> > **6. Another potential weakness is that all of the datasets are homophilous. I am not sure how will this perform on heterophilous graphs.
> 7. Another weakness is that it is not clear how this approach is directly applicable to heterogeneous graphs (which authors mention as well).**
>
> Thank you for the suggestion, we have currently experimented with datasets from different sources and very different scale sizes for which our method seems to be fairly accurate and scalable as well. We believe that the performance trend should be similar for heterophilous graphs and we will add experiments for these additional datasets in the final version owing to the short rebuttal timeline. For heterogeneous graphs, we agree that the task of clustering will have to be defined differently altogether, maybe at an edge level and the contrastive formulation can be tweaked accordingly based on the type of connections, however this would need more investigation which is why we foresee this as an immediate future work.
>
>
> > **8. Is there an ablation that uses only the normalized adjacency/diffusion matrix in the GCN. I didn’t find these in the paper (or in the appendix).**
>
> We attach an ablation for the same on different datasets here and report the average clustering NMI :
>
> | Dataset   | S3GC - AX | S3GC-SX | S3GC - both AX and SX (reported in the paper) |
> | ----------- | ----------- |----------- | ----------- |
> | Cora           | 0.554      | 0.565    |0.588  |
> | Citeseer     |  0.414| 0.398    | 0.441       |
> | Pubmed     | 0.279      | 0.330      | 0.333       |
> | ogbn-arxiv    |  0.460       | 0.454       |0.463       |
> | ogbn-products  |  0.531       | 0.513       |0.536      |
>
>
> > **“This paper refers to DGI as SOTA (page 6 lines 233-235, 273-274). Is that true? Perhaps the authors meant to say DGI style methods? MVGRL has been shown to work better and that is validated in the experiments as well (Table 3). Thus, I think Table 1 should have results with MVGRL. I understand authors have concerns that the reported values are higher in MVGRL (line 317-318) but I think it is good to add for completeness..”**
>
> Thank you for the suggestion. We meant to say that both DGI and DGI based methods are state-of-the-art that are also scalable. Among these in terms of both implementation and methodology, DGI itself is highly scalable, however some other DGI style methods like MVGRL aren’t scalable (we have detailed the reason for this in the response above), which is why for the comparisons in Table-1, we compare only with the other scalable and state-of-the-art methods, which in our case are Node2Vec and DGI. Others don’t scale to graphs of order of Millions of nodes which is often a practical requirement in the real world setting, as we discuss in the paper. However for completeness, we will also add MVGRL to Table 1 for comparison in the final version.
>
>
> ------------------------------------------------
>
> We hope that the rebuttal clarifies questions raised by the reviewer. We would be very happy to discuss any further questions about the work, and would really appreciate an appropriate increase in score if reviewers’ concerns are adequately addressed to facilitate acceptance of the paper.

---

### Comment · Area_Chair_A6b3 · 2022-08-06
**AC comment to authors**

After reading the reviews and the paper, I was not able to identify whether the proposed method outputs positional embeddings or structural embeddings. These have very different clustering properties (see Figures 3 and 4 of Srinivasan et al., 2020 to understand the differences). The first would cluster nodes based on the positions in the graph, while the second would cluster nodes based on their structural similarity. It all boils down to how the algorithms react to symmetric graphs (https://en.wikipedia.org/wiki/Symmetric_graph). Looking at the algorithm, I looks like it outputs structural embeddings (as opposed to node2vec, which outputs positional embeddings). But I am not certain. Could the authors please clarify?

Srinivasan, Balasubramaniam, and Bruno Ribeiro. "On the equivalence between positional node embeddings and structural graph representations." ICLR 2020. https://arxiv.org/pdf/1910.00452.pdf

---

> ### Author Response · Authors · 2022-08-08
> **Response to AC Question**
>
> Thanks for the interesting question. Our method is also doing positional encoding similar to node2vec, as we are forming positives/negatives in our contrastive learning formulation based on some kind of random walk. But naturally structure of the graph also plays a role in the encoding. Hope that clarifies, happy to further engage in the discussion.

---

### Meta-Review · Area_Chair_A6b3 · 2022-08-27

**Recommendation:** Accept
**Confidence:** Less certain

**Metareview:**

TL;DR: Accept is there is room.

The paper proposes a Scalable Self-Supervised method that uses graph neural networks (1 layered graph convolutional network to encode feature and structural information), diffusion augmentations, and contrastive learning for graph clustering.  The work is more applied, the baselines are somewhat weak (but these are the ones that scale), and overall there is nothing ground-breaking about this work. Generally NeurIPS does not accept this type of work but it could be interesting to part of the community focused on scalability.

During rebuttal almost all reviewer concerns were addressed. The authors did a good job showing with their experiments.

It worries me that the authors answered the questions but did not make significant changes to their draft (despite saying they would do that). I was inclined to reject the paper because of the absence of updates but I will trust the authors' word this time. But if the discussed changes are not implemented there should be consequences.

"Our method is also doing positional encoding similar to node2vec, as we are forming positives/negatives in our contrastive learning formulation based on some kind of random walk." => I read it again and I highly doubt this statement. Contrastive learning does not necessarily create positional representations. Please clearly state the type of embedding and add a proof to your paper (it can be added in the appendix).

**Award:**

No

---

### Decision · Program_Chairs · 2022-09-14

Accept